# Distinguishing Learning Rules with Brain Machine Interfaces

**Jacob P. Portes**
Center for Theoretical Neuroscience
Columbia University
j.portes@columbia.edu

**Christian Schmid**
Institute of Neuroscience
University of Oregon
cschmid9@uoregon.edu

**James M. Murray**[*]
Institute of Neuroscience
University of Oregon
jmurray9@uoregon.edu

## Abstract

Despite extensive theoretical work on biologically plausible learning rules, clear evidence about whether and how such rules are implemented in the brain has been difficult to obtain. We consider biologically plausible supervised- and reinforcement-learning rules and ask whether changes in network activity during learning can be used to determine which learning rule is being used. Supervised learning requires a credit-assignment model estimating the mapping from neural activity to behavior, and, in a biological organism, this model will inevitably be an imperfect approximation of the ideal mapping, leading to a bias in the direction of the weight updates relative to the true gradient. Reinforcement learning, on the other hand, requires no credit-assignment model and tends to make weight updates following the true gradient direction. We derive a metric to distinguish between learning rules by observing changes in the network activity during learning, given that the mapping from brain to behavior is known by the experimenter. Because brain-machine interface (BMI) experiments allow for precise knowledge of this mapping, we model a cursor-control BMI task using recurrent neural networks, showing that learning rules can be distinguished in simulated experiments using only observations that a neuroscience experimenter would plausibly have access to.

## 1 Introduction

In order to update synaptic weights effectively during learning, a biological or artificial neural network must solve the problem of credit assignment. That is, a neuron must infer whether it should increase or decrease its activity in order to decrease the error in the behavior that it is driving. Supervised learning (SL) provides one approach for solving the credit assignment problem by endowing the synaptic learning rule with a *credit assignment mapping*, i.e. an estimate of how each neuron's activity affects the behavioral readout. Indeed, gradient-based methods such as backpropagation for feedforward networks and backpropagation through time for recurrent networks do exactly this by multiplying a (possibly multi-dimensional) readout error by the downstream readout weights in order to update hidden-layer synaptic weights. However, because neurons in a biological neural network likely lack complete information about their downstream projections, ideal credit assignment mappings such as those used in gradient-based methods are biologically implausible–an issue known as the weight transport problem [1, 2, 3]. Recent work has shown that effective supervised learning

---

[*]corresponding author

in both feedforward networks and RNNs can still take place as long as there is a positive overlap between the true readout weights and the credit assignment matrix used for learning, providing one possible solution to this problem [4, 3, 5, 6, 7].

Reinforcement learning (RL) is another approach for solving the credit assignment problem. Node perturbation algorithms correlate noise injections with scalar reward signals in order to update network weights [8, 9, 10]. Such algorithms avoid the weight transport problem, as the update rule does not explicitly require a credit assignment matrix mapping the vector error back to the recurrent weights. While the weight updates for algorithms in this family tend to be noisy, the policy-gradient theorem guarantees that they tend to follow the true gradient of the objective function (Appendix B.2, [8, 11]).

Brain-machine interfaces (BMI) provide an ideal paradigm for investigating questions about learning and credit assignment in real brains, as they enable the experimenter to define the mapping (the "decoder") from neural activity onto behavior. For example, monkeys can be trained to control a cursor using a BMI in the motor cortex, and the behavior can be relearned following different manipulations of the BMI decoder in the course of a single day [12, 13, 14] or over the course of multiple days [15, 16, 17]. In the case of BMIs, the weight transport problem is particularly acute, since the readout weights mapping the neural activity onto the cursor position may be changed abruptly by the experimenter, and there is no plausible way for the agent to generate an instantaneous estimate of the new readout weights in order to then apply a learning rule. If an agent is to use a credit-assignment model at all in such a task, it must consist of a weight matrix that imperfectly approximates the readout mapping, i.e. a *biased* internal model of credit assignment. We refer to this as the *decoder alignment problem*.

The key insight in our work is that, under an SL rule, the agent will invariably perform credit assignment using a biased credit-assignment model, whereas no such model is required under a policy gradient-like RL rule. Using this insight, we develop a framework for modeling BMI experiments with recurrent neural networks (RNNs) using biologically plausible versions of SL and RL. We show that an imperfect credit-assignment model introduces a systematic bias in the weight updates, as well as the direction in which neural activity evolves during learning. Under the assumption that SL is biased but RL is not, this therefore provides a means by which to distinguish these different learning strategies. We derive a statistical metric for comparing observed changes in neural activity with the changes predicted assuming knowledge of the BMI decoder, and we show how this metric can be used to detect learning bias in our simulated experiments. While we focus on distinguishing biased SL from unbiased RL, the approach described here could be used more generally to distinguish different biased vs. unbiased learning rules—or even biased learning rules with two different types of bias—from one another.

**Related work.** While there has recently been a renaissance of research on biologically plausible learning rules [4, 10, 6, 18, 5, 3], there has been relatively little work on how such proposals might be experimentally verified in neural experiments. Lim et al. [19] inferred hyperparameters of a Hebbian learning rule based on neural activity from a visual cortical area in monkeys. Their work, however, assumed a single class of learning rules rather than trying to distinguish between multiple classes. Nayebi et al. [20] showed that different optimizers and hyperparameters used to train convolutional neural networks can in principle be distinguished by a classifier based on both weight changes and neural activations. However, the classifier was trained using ground-truth data about the identity of the learning rule, which is the very thing that we aim to infer in our model. Kepple et al. [21] proposed a curriculum-based method for discerning the objective being optimized with SL and showed via RNN modeling that different objectives lead to distinct task-acquisition timescales. While their approach analyzed evidence accumulation and decision tasks and focused on distinguishing different objectives, we focus primarily on continuous control tasks and on distinguishing learning rules. While Feulner and Clopath [22] simulated BMI learning with RNNs to compare learning that required changes within vs. outside of the intrinsic manifold of neural activity, and Feulner et al. [23] used a similar approach to analyze how learning at upstream or recurrent synapses affected the covariance of neural activity, neither of these studies attempted to distinguish learning rules. To our knowledge, our study is the first to propose a framework for distinguishing between learning rules using BMIs.

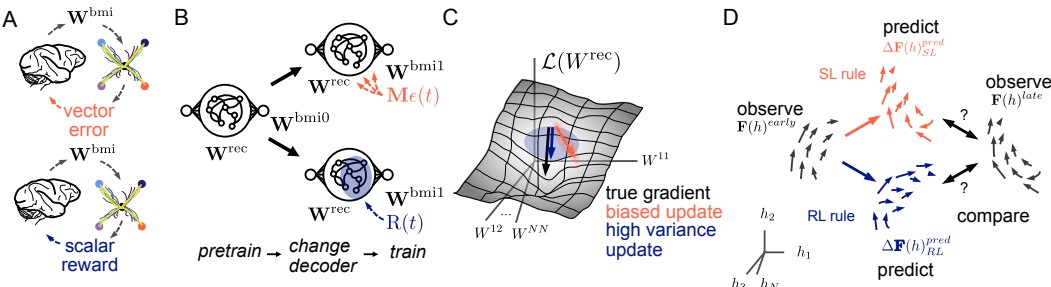

Figure 1: Approach. (A) We model a brain-machine interface experiment in which a monkey learns to move a cursor on a screen using either supervised learning (SL, top) or reinforcement learning (RL, bottom). (B) We pretrain an RNN to perform a center-out reach task with a fixed decoder $\mathbf{W}^{\mathrm{bmi0}}$, switch the decoder to $\mathbf{W}^{\mathrm{bmi1}}$, then train the recurrent weights $\mathbf{W}^{\mathrm{rec}}$ again with either SL or RL. (C) SL with a biased credit assignment mapping will lead to biased recurrent weight updates, whereas RL leads to noisy but unbiased weight updates. (D) By inferring the neural activity flow fields from observed data and comparing the observed change during learning with the predicted changes assuming SL or RL, we infer which learning rule was used to train the RNN.

## 2 Theoretical approach

The overall approach for our work is shown schematically in Fig. 1. We train RNNs to perform a BMI-inspired cursor-control task using different learning rules, with the goal of analyzing the observed activity to infer the learning rule that was used to train the network. Inspired by numerous BMI experiments [12, 24, 16], we pre-train the RNN using a particular choice of decoder weights, then require the RNN to relearn the task using a new set of decoder weights. Our main result is to predict the different ways in which the RNN activity is expected to evolve through learning, then to compare these predictions with the observed changes in the activity in order to infer the learning rule. In the sections that follow, we first introduce the SL and RL update rules that are used for training our networks. We then show that they lead to distinct predictions about how the RNN activity is expected to evolve with training. The remainder of the paper presents empirical results showing that the learning rules can be inferred by observing the network activity, first for a relatively simple case and then under more generalized assumptions.

While our main results can be obtained using either a feedforward or recurrent circuit model, we mostly focus on the latter case because BMI experiments typically involve control tasks driven by time-varying activity in motor cortex, for which an RNN model is more appropriate. The feedforward version of our results is provided in Appendix A.

### 2.1 Modeling BMI learning with RNNs

We model motor cortex as a vanilla RNN characterized by input weights $\mathbf{W}^{\mathrm{in}}$, recurrent weights $\mathbf{W}^{\mathrm{rec}}$, feedback weights $\mathbf{W}^{\mathrm{fb}}$, and BMI decoder weights $\mathbf{W}^{\mathrm{bmi}}$. Since the experimenter has complete control over the decoder in a BMI experiment, we treat $\mathbf{W}^{\mathrm{bmi}}$ as fixed and assume that learning occurs in recurrent weights $\mathbf{W}^{\mathrm{rec}}$ alone. The activity of the recurrent units $\mathbf{h}^t$ is given by

$$h_i^t = \Big(1 - \frac{1}{\tau}\Big)h_i^{t-1} + \frac{1}{\tau}\phi\Big(\sum_j W_{ij}^{\mathrm{rec}}h_j^{t-1} + \sum_j W_{ij}^{\mathrm{in}}x_j^t + \sum_j W_{ij}^{\mathrm{fb}}y_j^{t-1}\Big) + \xi_i^t, \qquad (1)$$

where $\tau$ is the RNN time constant, $\phi(\cdot) = \tanh(\cdot)$ is a nonlinear activation function, $\mathbf{x}^t$ is an input signal, and $\xi_i^t \sim \mathcal{N}(0, \sigma_{\mathrm{rec}}^2)$ is isotropic noise added to the recurrent units. This activity is read out by $\mathbf{W}^{\mathrm{bmi}}$ to obtain the cursor position $y_k^t = \sum_i W_{ki}^{\mathrm{bmi}}h_i^t$. The error at each time step is defined as $\epsilon_k^t = y^*{}_k^t - y_k^t$, where $\mathbf{y}^{*t}$ is the target output for each time step. The loss to be minimized is then $L = \frac{1}{2T}\sum_{t=1}^{T}\sum_{k=1}^{N_y}(\epsilon_k^t)^2$ where $T$ is the trial duration and $N_y = 2$ is the number of readout dimensions.

**Supervised learning.** Standard supervised algorithms for RNNs such as Backpropagation Through Time [25] and Real-Time Recurrent Learning [26] are widely acknowledged to be biologically

implausible. The reasons for this are two-fold. First, both algorithms assume knowledge of the readout weights for purposes of credit assignment, which we referred to above as the weight transport problem. Second, updating a particular recurrent weight with one of these algorithms requires knowledge of all of the other weights in the network, which is not information that a biological neuron would likely have access to.

A recently proposed alternative, Random Feedback Local Online (RFLO) learning [5], is an approximate gradient-based algorithm with $\Delta \mathbf{W}^{\mathrm{rec}} \approx -\partial L / \partial \mathbf{W}^{\mathrm{rec}}$ that addresses these problems by (i) assuming that error information is projected back into the network with weights that imperfectly approximate the ideal credit assignment mapping, and (ii) dropping nonlocal terms from the update rule. The recurrent weight update with this learning rule is

$$\Delta W_{ij}^{\mathrm{rec}} = \eta \sum_{t=1}^{T} \left[ \mathbf{M} \boldsymbol{\epsilon}^t \right]_i p_{ij}^t, \qquad p_{ij}^t = \left( 1 - \frac{1}{\tau} \right) p_{ij}^{t-1} + \frac{1}{\tau} \phi'(u_i^t) h_j^{t-1}, \qquad (2)$$

with $u_i^t = \sum_k W_{ik}^{\mathrm{rec}} h_k^{t-1} + \sum_k W_{ik}^{\mathrm{in}} x_k^t + \sum_k W_{ik}^{\mathrm{fb}} y_k^t$, and $p_{ij}^t \approx \partial h_i^t / \partial W_{ij}^{\mathrm{rec}}$. Further details are provided in Appendix B.1. The learning rule that we implement differs slightly from RFLO in that the readout weights, as in a BMI experiment, are fixed during training rather than learned. In RFLO, learning of the readout weights was required in order to align the readout weights with the credit assignment matrix. Here we will instead assume that the credit assignment matrix $\mathbf{M}$ is fixed and partially aligned with $(\mathbf{W}^{\mathrm{bmi}})^\top$, so that the cosine similarity of the flattened matrices satisfies $\mathrm{sim}(\mathbf{M}, (\mathbf{W}^{\mathrm{bmi}})^\top) > 0$.

In (2), the credit assignment mapping $M_{ik}$ assigns responsibility for the readout error $\epsilon_k^t$ to neuron $i$. While the ideal mapping would be $\mathbf{M} = (\mathbf{W}^{\mathrm{bmi}})^\top$, learning can still be effective as long as there is a positive alignment between $\mathbf{M}$ and $(\mathbf{W}^{\mathrm{bmi}})^\top$ [5] (as shown previously in the case of feedforward networks [4]). In the case where they are not equal, as we presume they cannot be in a BMI experiment, the matrix $\mathbf{M}$ constitutes a *biased* credit-assignment mapping. Below, we show that this leads to a predictable bias in the change of the RNN activity as the task is learned.

Example output trajectories from an RNN trained on the cursor-control task with the SL rule described above are shown in Fig. 2A. The results in Fig. 2B show that learning the task is possible whenever there is partial alignment between $\mathbf{M}$ and $(\mathbf{W}^{\mathrm{bmi}})^\top$, and that faster learning occurs as this alignment increases.

**Reinforcement learning.** To train RNNs with RL, we use a node perturbation learning rule closely related to REINFORCE [8]. We first define a scalar reward $R^t = -|\boldsymbol{\epsilon}^t|^2$. Node perturbation seeks to increase this reward by trial and error using the noise appearing in the recurrent population activity in (1). If the noise to a given neuron causes the reward to increase, then the weights will be updated so that that neuron's input will be greater in subsequent trials. Including an eligibility trace in the learning rule further makes it possible to assign credit for changes in reward to noise events at previous time steps. Finally, we can subtract off a baseline from the reward so that learning is guided by a reward prediction error, which somewhat decreases the variance of the weight updates without introducing bias [11]. The resulting learning rule, which we derive using the policy gradient theorem [27, 11] in Appendix B.2, is

$$\Delta W_{ij}^{\mathrm{rec}} = \eta \sum_{t}^{T} (R^t - \bar{R}^t) q_{ij}^t, \qquad q_{ij}^t = \left( 1 - \frac{1}{\tau} \right) q_{ij}^{t-1} + \frac{1}{\tau} \xi_i^t \phi'(u_i^t) h_j^{t-1}. \qquad (3)$$

Very similar RL rules for RNNs have been previously proposed and studied in Refs. [28, 10]. Importantly for our purposes, the fact that this learning rule is derived using a policy gradient approach guarantees that, upon averaging over isotropic noise, the weight updates follow the true gradient of the loss function (in our case given by $L = -\frac{1}{2} \sum_t R^t$), making this an *unbiased* learning rule. Like (2), (3) does not have nonlocal terms. Example output trajectories and the learning curve from an RNN trained on the cursor-control task using (3) are shown in Fig. 2D,E.

## 2.2 Characterizing changes in neural activity with vector flow fields

Because observing changes in neural activity is much more straightforward than observing changes in synaptic weights in neuroscience experiments, we aim to identify experimentally observable signatures in the neural activity that might distinguish biased SL from unbiased RL. Because the

RNN activity consists of time-dependent trajectories, and changes in the weights will generally affect entire trajectories cumulatively from one time step to the next, calculating the change $\Delta\mathbf{h}^t$ due to an update $\Delta\mathbf{W}^{\text{rec}}$ is not entirely straightforward (though *cf.* the feedforward case described in Appendix A, where this can be done). A more useful approach is to quantify the learning-induced change in the vector flow field that shapes the RNN activity, where the flow field can be estimated empirically by observing neural activity trajectories. To do this, we linearize (1), assuming $\phi(u) = u$, to obtain an expression for a vector flow field function $\mathbf{F}(\mathbf{h})$ that maps activity $\mathbf{h}^t \longrightarrow \mathbf{h}^{t+1}$ and is defined for all points $\mathbf{h}$ in neural activity space:

$$\mathbf{F}(\mathbf{h}) = \frac{1}{\tau}\left[(\mathbf{W}^{\text{rec}} - \mathbb{I})\mathbf{h} + \mathbf{W}^{\text{in}}\mathbf{x} + \mathbf{W}^{\text{fb}}\mathbf{y}\right], \tag{4}$$

where, in our simulations, $\Delta t = 1$. Thus, from (1), we see that $\mathbf{F}(\mathbf{h}^t) \approx \mathbf{h}^{t+\Delta t} - \mathbf{h}^t$. If the recurrent weights change according to $\mathbf{W}^{\text{rec}} \to \mathbf{W}^{\text{rec}} + \Delta\mathbf{W}^{\text{rec}}$, then the flow field exhibits a corresponding change $\mathbf{F} \to \mathbf{F} + \Delta\mathbf{F}$, where $\Delta\mathbf{F}(\mathbf{h}) = \frac{1}{\tau}\Delta\mathbf{W}^{\text{rec}}\mathbf{h}$. This equation enables us to relate changes in $\mathbf{W}^{\text{rec}}$ to changes in neural activity (note that this assumes that plasticity is occurring in the recurrent weights $\mathbf{W}^{\text{rec}}$ alone, and that $\mathbf{W}^{\text{in}}$ and $\mathbf{W}^{\text{fb}}$ do not change). As illustrated in Fig. 1D, our approach will be to (i) predict the expected change in flow field $\Delta\mathbf{F}^{\text{pred}}(\mathbf{h})$ under either SL or RL, (ii) estimate $\Delta\mathbf{F}^{\text{obs}}(\mathbf{h})$ empirically using activity observed early and late in learning, and (iii) determine which learning rule was used to train the network by comparing the empirically observed vs. predicted flow field changes.

In order to predict the flow field change in the SL case for a point $\mathbf{h}$ in neural activity space, we predict the direction of weight change using an approximation of (2):

$$\Delta\mathbf{F}^{\text{pred}}_{\text{SL}}(\mathbf{h}) = \Delta\mathbf{W}^{\text{pred}}|_{\text{SL}}\mathbf{h}, \qquad \Delta W^{\text{pred}}_{ij}|_{\text{SL}} = \sum_{n\in\text{mid}}\sum_t\sum_k M_{ik}\epsilon^{n,t}_k h^{n,t}_j, \tag{5}$$

where $n$ refers to trials, and $n \in \text{mid}$ denotes that we sample from trials during the middle part of training, when most of the learning is presumed to take place. In the RL case, we similarly predict the direction of weight change using an approximation of (3):

$$\Delta\mathbf{F}^{\text{pred}}_{\text{RL}}(\mathbf{h}) = \Delta\mathbf{W}^{\text{pred}}|_{\text{RL}}\mathbf{h}, \qquad \Delta W^{\text{pred}}_{ij}|_{\text{RL}} = \sum_{n\in\text{mid}}\sum_t\sum_k W^{\text{bmi}}_{kl}\Sigma_{il}\epsilon^{n,t}_k h^{n,t}_j. \tag{6}$$

In this equation, $\boldsymbol{\Sigma}$ is the recurrent noise covariance, which, under conditions of isotropic noise, reduces to a prefactor $\sigma^2_{\text{rec}}\mathbb{I}$.

Importantly for our purposes, the predicted changes in flow field from (5) and (6) depend entirely on quantities that are known or can be estimated from experimental data, including $\mathbf{W}^{\text{bmi}}$, as well as samples of $\mathbf{h}^{n,t}$ and $\boldsymbol{\epsilon}^{n,t}$. As shown in Section 3.4 below, $\mathbf{M}$ can also be estimated from data. Because we are asking only about changes in the flow field rather than the flow field itself, these quantities do not depend explicitly on the recurrent connectivity $\mathbf{W}^{\text{rec}}$ itself, which is presumably unobtainable experimentally. The key distinction between (5) and (6) is that the change in flow field in the SL case depends on $\mathbf{M}$, while the change in flow field in the RL case depends on $(\mathbf{W}^{\text{bmi}})^\top$. We motivate (5) and (6) by computing $\langle\Delta\mathbf{W}^{\text{rec}}\rangle$ for each learning rule (averaging over noise) in Appendix B.1 and Appendix B.2, respectively. We provide empirical evidence that updates approximately follow a consistent direction for each rule in Appendix C.5.

Next, we wish to estimate the empirical change in the flow field given neural activity observed at different points in learning. The approach is somewhat simplified if we assume that the trials during which learning takes place are preceded and followed by "early" and "late" blocks of trials, in which the performance is observed but the weights are not updated. The assumption that the amount of learning during these early and late blocks is negligible is justified in the limit where the learning rate is small and the number of training trials is large. (Evidence that our main results do not depend strongly on this assumption is given in Appendix A, where learning was allowed to take place during the early and late blocks.) Then, from (4), we define the observed flow field from time step $t$ in trial $n$ as either $\mathbf{F}^{\text{early}}(\mathbf{h}) = (\mathbf{A}^{\text{early}} - \mathbb{I})\mathbf{h}$ or $\mathbf{F}^{\text{late}}(\mathbf{h}) = (\mathbf{A}^{\text{late}} - \mathbb{I})\mathbf{h}$, where $\mathbf{A}^{\text{early}}$ is the least-squares solution to the autoregression equation $\mathbf{h}^{n,t+1} = \mathbf{A}^{\text{early}}\mathbf{h}^{n,t}$ for all time steps $t$ in all early trials $n$, and $\mathbf{A}^{\text{late}}$ is the corresponding solution for late trials. The empirical change in the flow field due to learning for any point $\mathbf{h}$ is then given by $\Delta\mathbf{F}^{\text{obs}}(\mathbf{h}) = \mathbf{F}^{\text{late}}(\mathbf{h}) - \mathbf{F}^{\text{early}}(\mathbf{h})$.

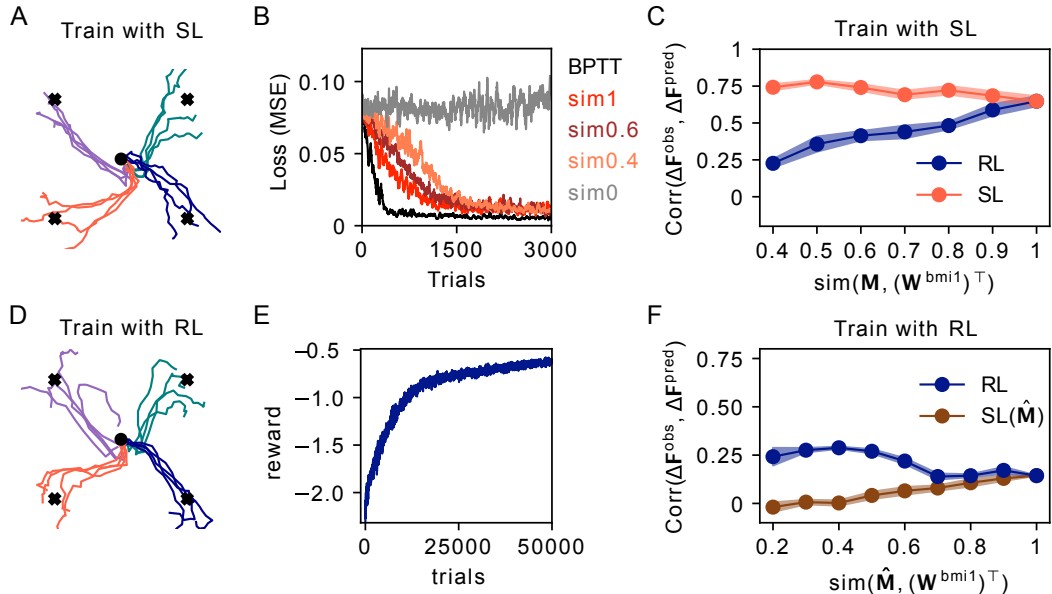

Figure 2: Distinguishing learning rules in a trained network. (A) Example output traces from an RNN trained with SL to move a cursor to one of four targets. (B) When trained with SL, the similarity of the credit assignment mapping with the readout weights determines the speed of training (red curves). Learning in an RNN trained with backpropagation through time is shown for comparison (black curve). (C) For an RNN trained with SL, the similarity of the observed flow field change with the predicted change assuming either SL (red) or RL (blue). (D) Example output traces from an RNN trained with RL to move a cursor to one of four targets. (E) Learning curve for an RNN trained with RL. (F) For an RNN trained with RL, the similarity of the observed flow field change with the predicted change assuming either SL with a randomly sampled $\hat{\mathbf{M}}$ (brown) or RL (blue).

Having defined the observed and predicted changes in the flow fields for arbitrary $\mathbf{h}$, we can define the correlation between these quantities as

$$\text{Corr}(\Delta\mathbf{F}^{\text{obs}}, \Delta\mathbf{F}^{\text{pred}}) = \frac{1}{N_{\text{trials}}T} \sum_{n=1}^{N_{\text{trials}}} \sum_{t=1}^{T} \frac{\Delta\mathbf{F}^{\text{obs}}(\mathbf{h}^{n,t}) \cdot \Delta\mathbf{F}^{\text{pred}}(\mathbf{h}^{n,t})}{|\Delta\mathbf{F}^{\text{obs}}(\mathbf{h}^{n,t})| \, |\Delta\mathbf{F}^{\text{pred}}(\mathbf{h}^{n,t})|}. \tag{7}$$

We refer to this as the flow field change correlation (FFCC). One important point to note is that, because this quantity is computed only at observed values of $\mathbf{h}^{n,t}$, it is not the case that the flow field needs to be estimated everywhere in the high-dimensional space of neural activity. If the activity tends to occupy a low-dimensional manifold within this space, then the flow field needs to be determined only within this manifold. Another point to note is that this metric is independent of the magnitudes of $\Delta\mathbf{F}^{\text{obs}}$ and $\Delta\mathbf{F}^{\text{pred}}$, so that it does not depend on overall prefactors such as the learning rates and network time constant. In the simulations that follow, the correlation metric (7) will be computed separately under our two hypothesized learning rules, i.e. with $\Delta\mathbf{F}^{\text{pred}} = \Delta\mathbf{F}^{\text{pred}}_{\text{SL}}$ and with $\Delta\mathbf{F}^{\text{pred}} = \Delta\mathbf{F}^{\text{pred}}_{\text{RL}}$. While the absolute magnitudes of this quantity are not informative on their own, we show below that the relative values can be compared to infer which of the two learning rules is more likely to have generated the data.

## 3 Simulation results

In this section, we use the theory outlined above to show that the SL and RL rules we consider can be distinguished in simulations, first in a basic case with simplifying assumptions, and then in more realistic cases where these assumptions are relaxed.

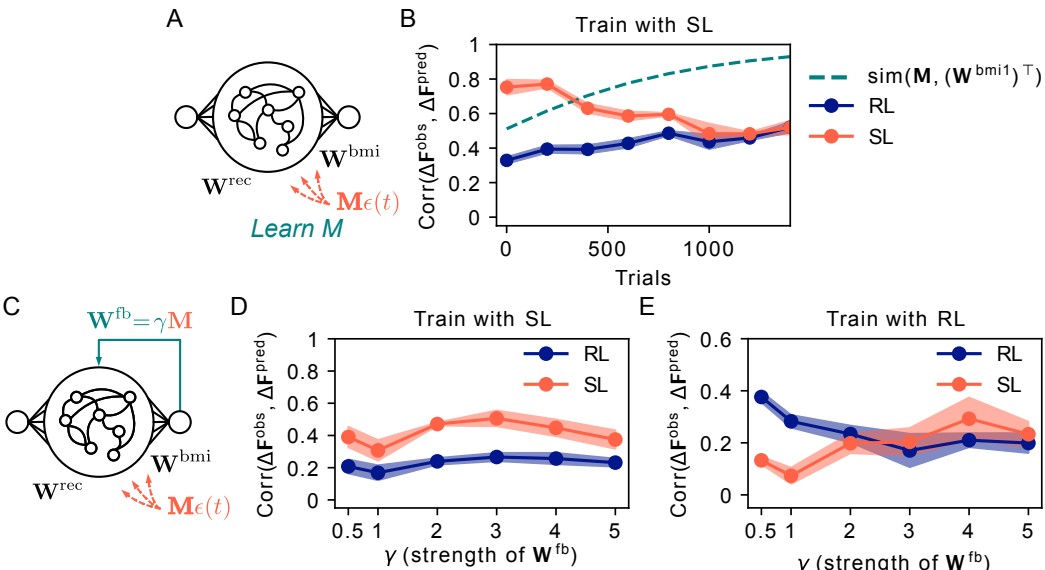

Figure 3: Learning the credit assignment mapping and driving with visual feedback. (A) In an RNN trained with SL, the credit assignment mapping $\mathbf{M}$ is learned online during training. (B) The similarity of $\mathbf{M}$ to $(\mathbf{W}^{bmi})^\top$ increases with training due to learning of $\mathbf{M}$ (green). The flow field metric correctly identifies SL as the learning rule early in training when $\mathbf{M}$ and $\mathbf{W}^{bmi}$ are dissimilar, but not later in training when they are very similar. (C) An RNN (now with fixed $\mathbf{M}$) is additionally driven by a feedback signal via weights $\mathbf{W}^{fb} \propto \mathbf{M}$. (D) Comparison between predicted and observed changes in the activity flow field as a function of feedback gain for an RNN trained with SL. (E) Comparison between predicted and observed changes in the activity flow field as a function of feedback gain for an RNN trained with RL.

## 3.1 Distinct neural signatures of biased SL and unbiased RL

In order to emulate monkey BMI experiments (e.g., [12, 16, 15, 13, 14]), we first pretrain the recurrent weights $\mathbf{W}^{rec}$ of an RNN with fixed, random readout weights $\mathbf{W}^{bmi0}$ to perform a center-out cursor-control task, in which a cursor must be moved to one of four target locations specified by the input to the RNN. We then change the BMI decoder to $\mathbf{W}^{bmi1}$, make identical copies of the network, and train one with SL using credit-assignment mapping $\mathbf{M}$ and the other with RL. For these simulations, shown in Fig. 2, we make the following simplifying assumptions (to be generalized in later sections): (i) the credit assignment mapping $\mathbf{M}$ is fixed and known; (ii) the feedback weights $\mathbf{W}^{fb}$ are zero; and (iii) the noise $\boldsymbol{\xi}^t \sim \mathcal{N}(0, \sigma_{rec}^2 \mathbb{I})$ is isotropic.

For the networks trained with biased SL and $\mathrm{sim}(\mathbf{W}^{bmi0}, \mathbf{W}^{bmi1}) = 0.5$, we find that a comparison of the observed and predicted flow field changes correctly detects the bias (Fig. 2C). As we vary the alignment between $\mathbf{M}$ and $(\mathbf{W}^{bmi1})^\top$, we find that the FFCC correctly detects the bias present during learning whenever these two matrices are sufficiently distinct. For the networks trained with RL, we find that the change in activity is aligned more closely to the prediction using true decoder $\mathbf{W}^{bmi1}$ than when using a random matrix $\hat{\mathbf{M}}$ that is partially aligned to $\mathbf{W}^{bmi1}$. As we increase the alignment $\mathrm{sim}(\hat{\mathbf{M}}, (\mathbf{W}^{bmi1})^\top)$, this distinction is less discernible (Fig. 2F). We show in Appendix C.1 that these results are robust to changes in hyperparameters.

In the empirical results that follow, we will generalize the assumptions made thus far and show that there is a significant range of parameters where the results still hold. In particular, we will consider the roles played by (i) estimation and/or learning of the credit-assignment model $\mathbf{M}$, (ii) driving feedback via the weights $\mathbf{W}^{fb}$, and (iii) non-isotropic noise.

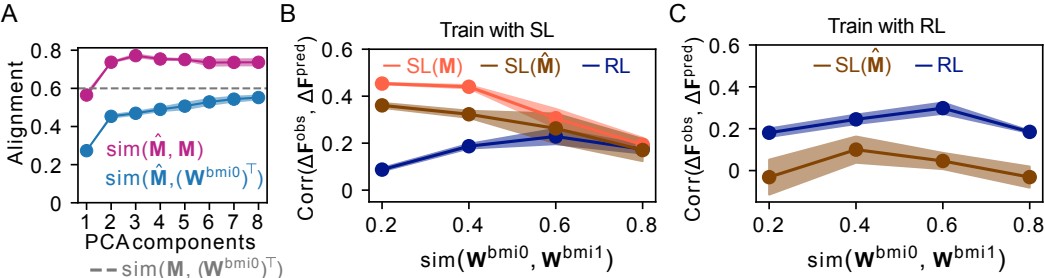

Figure 4: Estimating the credit assignment mapping. (A) The similarities between the estimated $\hat{\mathbf{M}}$ and the true $\mathbf{M}$ (magenta) and between $\hat{\mathbf{M}}$ and $(\mathbf{W}^{\text{bmi0}})^{\top}$ (blue) as a function of the number of principal components used in the regression in an RNN trained with SL. (B) Comparison between predicted and observed changes in the activity flow field as a function of old vs. new decoder alignment for an RNN trained with SL. Model predictions assume SL using the true credit assignment mapping $\mathbf{M}$ (red), SL using the $\hat{\mathbf{M}}$ estimated from pretraining (brown), or RL (blue). (C) Same as (B), for an RNN trained with RL (and where $\hat{\mathbf{M}}$ is estimated from pretraining).

## 3.2 Learning the credit assignment mapping

In order to optimize learning in a supervised setting, a learner's credit assignment mapping could improve over time to align more closely with the decoder (Fig. 3A). To investigate the effect of such learning in our model, we applied our analysis to an RNN in which recurrent weights $\mathbf{W}^{\text{rec}}$ were trained with SL, with a biased credit assignment mapping $\mathbf{M}$ that slowly improved via *weight mirroring*, a perturbation-based algorithm for learning a credit assignment matrix [6, 18]. In our simulations (Fig. 3B), the alignment between $\mathbf{M}$ and $(\mathbf{W}^{\text{bmi1}})^{\top}$ is initially set at $0.5$, then slowly increases as $\mathbf{M}$ is learned simultaneously along with the recurrent weights $\mathbf{W}^{\text{rec}}$. Instead of predicting the change in flow field before and after learning, we now apply our analysis over discrete windows during learning. As expected, activity changes are biased—and hence the SL and RL learning rules are distinguishable—early in learning. However, these differences disappear as $\mathbf{M}$ approaches $(\mathbf{W}^{\text{bmi1}})^{\top}$ later in learning.

## 3.3 Incorporating driving feedback

So far, we have not included *driving* feedback, which can strongly influence the dynamics of population activity. We next included fixed driving feedback weights $\mathbf{W}^{\text{fb}} = \gamma \mathbf{M}$ and varied the strength of these weights by a scalar factor $\gamma$ (Fig. 3C). $\mathbf{M}$ was randomly generated such that $\text{sim}(\mathbf{M}, (\mathbf{W}^{\text{bmi1}})^{\top}) = 0.5$. Note that the networks trained with SL therefore used the same mapping $\mathbf{M}$ for both the learning rule and for the driving feedback. We found that bias in an RNN trained with SL is correctly identified for a wide range of $\gamma$ (Fig. 3D). For RNNs trained with RL, we found that the correct learning rule is identified for weak feedback, but that the FFCC metric fails for strong driving feedback (Fig. 3E). This is likely because neural activity in this limit is dominated by the contribution from $\mathbf{W}^{\text{fb}}$, which is fully aligned with $\mathbf{M}$ and less aligned to $(\mathbf{W}^{\text{bmi1}})^{\top}$.

## 3.4 Estimating the credit assignment mapping from observed activity

Our approach thus far estimates the extent to which the activity updates follow either the direction of the gradient defined by the decoder weights or the (possibly biased) gradient defined by the credit assignment mapping. This estimation assumes knowledge of the credit assignment mapping $\mathbf{M}$. In a neuroscience experiment, however, this quantity would not be known *a priori* and would need to be estimated. To obtain an estimate $\hat{\mathbf{M}}$ of $\mathbf{M}$, we follow an approach previously used in experiments, in which dimensionality reduction is first applied to the neural data, followed by linear regression to relate the network activity to the cursor movements [12, 16, 22]. (A possible alternative approach could be to use the Internal Model Estimation framework proposed by Golub et al. [29].) More specifically, we (i) pretrain the RNN with decoder $\mathbf{W}^{\text{bmi0}}$ and biased credit assignment mapping $\mathbf{M}$ (where $\text{sim}(\mathbf{M}, (\mathbf{W}^{\text{bmi0}})^{\top}) = 0.6$), (ii) observe the activity and cursor trajectories produced by the trained network, (iii) perform principal component analysis on the network activity and relate

the principal components to the cursor positions with linear regression, and (iv) use the regression coefficients to define the estimated credit assignment mapping $\hat{\mathbf{M}} = \mathbf{DC}$, where $\mathbf{C}$ is the matrix of principal-component eigenvectors and $\mathbf{D}$ is the matrix of regression coefficients. Fig. 4A shows that the $\hat{\mathbf{M}}$ obtained in this way is similar to the true $\mathbf{M}$ used to train the RNN. It is important to note for our aim of distinguishing learning rules that $\hat{\mathbf{M}}$ is more similar to $\mathbf{M}$ than to $\mathbf{W}^{\mathrm{bmi0}}$ and correctly identifies the alignment between $\mathbf{M}$ and $(\mathbf{W}^{\mathrm{bmi0}})^\top$. We also see from Fig. 4A that this result depends only weakly on the number of principal components used to describe the network activity.

After estimating $\hat{\mathbf{M}}$, we changed the decoder to $\mathbf{W}^{\mathrm{bmi1}}$ and trained the network to proficiency with either SL (using $\mathbf{M}$ in the learning update) or RL (which doesn't use $\mathbf{M}$). For an RNN trained with SL (Fig. 4B), the correct learning rule is identified using either the true $\mathbf{M}$ (red curve) or the estimated $\hat{\mathbf{M}}$ (brown curve) for low and intermediate degrees of similarity between $\mathbf{W}^{\mathrm{bmi0}}$ and $\mathbf{W}^{\mathrm{bmi1}}$. For an RNN trained with RL (Fig. 4C), correct identification of the learning rule does not depend on the similarity between $\mathbf{W}^{\mathrm{bmi0}}$ and $\mathbf{W}^{\mathrm{bmi1}}$. Together, the above results show that, for a relatively wide range of parameters, the credit assignment mapping can be estimated from observed data, and that this estimate is sufficient to distinguish between learning rules used to train different networks.

### 3.5   RL with non-isotropic noise

In the theoretical and empirical results above, we have assumed that the noise covariance is isotropic. If we instead allow for non-isotropic recurrent noise covariance $\boldsymbol{\Sigma}$, we find that this covariance appears in the expected RL weight update (Appendix B.2). As in feedforward networks [30], this introduces a *bias* in the weight updates, so that the noise-averaged weight updates no longer follow the true gradient. Given that RL is no longer unbiased in this case, we thus asked under what conditions we might still be able to distinguish biased SL from RL.

We first established that the RNNs can learn a cursor-control task with non-isotropic noise via either SL (trivially) or RL (as long as some of the noise is in the subspace of the decoder). To do this, we pretrained networks with $\mathbf{W}^{\mathrm{bmi0}}$, changed the decoder to $\mathbf{W}^{\mathrm{bmi1}}$, and then trained using either SL or RL. In the SL case, we chose a biased credit mapping $\mathbf{M}$ that has partial overlap with the new decoder, with $\mathrm{sim}(\mathbf{M}, (\mathbf{W}^{\mathrm{bmi1}})^\top) = 0.6$. The noise covariance was chosen to be $d$-dimensional, with $\mathrm{rank}(\boldsymbol{\Sigma}) = d$, and isotropic within those dimensions. The first two of these dimensions were selected to lie in the subspace spanned by $\mathbf{M}$ for the SL-trained RNNs or by $\mathbf{W}^{\mathrm{bmi1}}$ for the RL-trained RNNs. Other components of $\boldsymbol{\Sigma}$ were added in random dimensions orthogonal to this subspace and to one another.

The results in Fig. 5 show that, when training with SL, we cannot reliably distinguish the learning rule used to train the network when the noise is very low-dimensional (Fig. 5B). When training with RL, however, the learning rule can be distinguished even with low-dimensional noise (Fig. 5C). As the noise dimensionality increases, the learning rules become more easily distinguishable in both cases. This remains true in the RL case even if we do not presume to know the true noise covariance matrix, but instead use a naive assumption of isotropic noise in order to generate the predictions (dark blue lines in Fig. 5B,C).

## 4   Discussion

In this work, we have proposed a method for distinguishing biased SL from RL under the assumption that the mapping from neural activity to behavior is known, as in a BMI experiment. More generally, the approach could be used to distinguish between different candidate learning rules with different degrees or types of bias. We have illustrated the approach by modeling a BMI cursor-control experiment in motor cortex, where there is ample experimental evidence for plasticity during motor learning [31]. Both SL and RL are plausible candidates for how motor learning might be implemented in the brain, either within motor cortex itself or within extended motor circuits involving motor cortex together with the cerebellum [32, 33, 34] or basal ganglia [35, 36].

One possible limitation of our approach is that the number of learning rules that might conceivably be implemented in recurrent circuits in the brain is much greater than the two that we have focused on here. Nevertheless, we view the SL and RL rules considered above as representative examples of two fundamental classes of learning rules: one in which learning is based on a model for credit

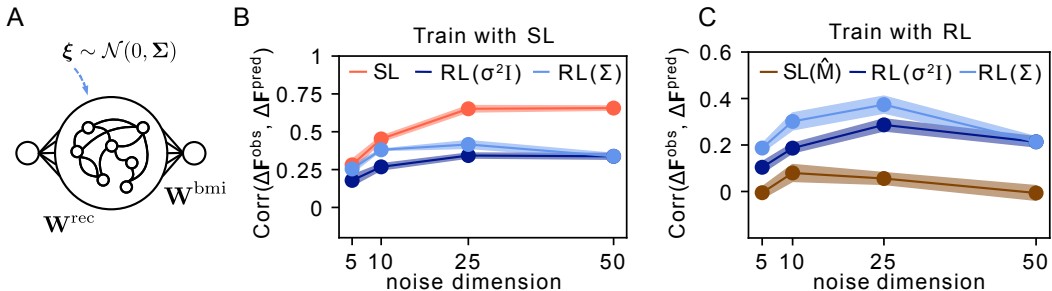

Figure 5: Distinguishing learning rules with non-isotropic noise. (A) Correlated noise with covariance matrix $\Sigma$ is injected into the recurrent units of the network during training and testing. (B) For RNNs trained with SL, the similarities between the observed flow field change and the predicted flow field change assuming SL (red), RL using the true noise covariance (light blue), or RL using a naive isotropic estimate of the noise covariance (dark blue). (C) Same as (B), but for RNNs trained with RL. Brown line is correlation of observed flow field change and predicted flow field change assuming SL with random $\hat{\mathbf{M}}$, where $\text{sim}(\hat{\mathbf{M}}, (\mathbf{W}^{\text{bmi1}})^{\top}) = 0.6$.

assignment, where this model is necessarily imperfect, and another which uses no such model. While we cannot test every learning algorithm from these classes, we conjecture that the basic fact that allows us to distinguish our SL and RL rules—namely, that the expected flow field change depends on a credit-assignment matrix for model-based rules but not for non-model-based rules—is a fairly generic feature of these two classes. To support this, we show in Appendix C.2 that our main results hold for a different version of the biased learning rule that we have studied.

While we have shown that different learning rules are in principle distinguishable in simulation, there will be challenges in extending this analysis to real data. Reliable estimation of the FFCC metric might require large populations of neurons recorded for long periods of time. Other potential challenges that will need to be addressed include accounting for the possibility that multiple types of learning might be occurring simultaneously at different locations, as well as considering cognitive mechanisms such as attention and contextual inference, which have not been included in this model. Finally, there are possible ambiguities to consider in mapping our model onto the brain's motor circuitry. While the most straightforward interpretation is that the RNN in our model corresponds to motor cortex, a more elaborate interpretation could be made by noting that both the cerebellum and the basal ganglia have well-established roles in motor learning [37] and are likely to play roles in BMI learning as well [36, 38].

## Acknowledgments and Disclosure of Funding

The authors would like to acknowledge Larry Abbott, Andrew Zimnik, Laureline Logiaco, Kaushik Lakshminarasimhan, and Vivek Athalye for fruitful discussions. Thanks to Amin Nejatbakhsh, David Clark, and Danil Tyulmankov for feedback on the manuscript. Support for this work was provided by NIH-NINDS (R00NS114194).

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
