# A  Distinguishing supervised learning from reinforcement learning in a feedforward model

In order to illustrate the main idea from our paper in a simplified context, we show in this section how observed hidden-layer activity in a linear feedforward network can be used to infer the learning rule that is used to train the network. Consider the simple feedforward network shown in Fig. S1. In this network, random binary input patterns $\mathbf{x}^t$, where $x_i^t \in \{-1, 1\}$ and $t = 1, \ldots, T$, are projected onto a hidden layer $\mathbf{h}^t = \mathbf{W}\mathbf{x}^t + \boldsymbol{\xi}^t$, where $\boldsymbol{\xi}^t \sim \mathcal{N}(0, \boldsymbol{\Sigma})$ is noise injected into the network. The readout is then given by $\mathbf{y}^t = \mathbf{W}^{\mathrm{bmi}}\mathbf{h}$, and the goal of the network is to minimize $L = \sum_t L_t = \frac{1}{T} \sum_t |\mathbf{y}^{*t} - \mathbf{y}^t|^2$, where $\mathbf{y}^*$ are randomly chosen target patterns with $y_i^* \sim \mathcal{N}(0, 1)$.

With $\mathbf{W}^{\mathrm{bmi}}$ fixed, we can train the network using either supervised learning (SL) or reinforcement learning (RL). In the case of SL, the learning rule performs credit assignment using a model $\mathbf{M}$ that approximates the ideal credit assignment model $(\mathbf{W}^{\mathrm{bmi}})^\top$ to project the error $\boldsymbol{\epsilon}^t = \mathbf{y}^{*t} - \mathbf{y}^t$ back to the hidden layer (Fig. S1A), giving the following weight update:

$$\Delta W_{ij}^{\mathrm{SL}} = \eta_{\mathrm{SL}} \sum_t [\mathbf{M}\boldsymbol{\epsilon}^t]_i x_j^t. \tag{8}$$

In the case where $\mathbf{M} = (\mathbf{W}^{\mathrm{bmi}})^\top$, this rule would be implementing gradient descent. In the case where $\mathbf{M} \neq (\mathbf{W}^{\mathrm{bmi}})^\top$ but the two matrices have positive alignment, it instead implements a biased version of gradient descent. This is similar to learning with Feedback Alignment [4], except that here we do not assume that the readout weights are being learned.

An alternative learning algorithm is policy gradient learning [8], which gives the following update equation:

$$\Delta W_{ij}^{\mathrm{RL}} = \eta_{\mathrm{RL}} \sum_t (R^t - \bar{R}^t)\xi_i^t x_j^t, \tag{9}$$

where we define the reward as $R^t = -L_t$, and we subtract off the baseline $\bar{R}^t = \langle L_t \rangle_\xi$.

Our goal will be to train the network to minimize the loss using either the SL or RL learning rule, then, assuming $\mathbf{W}^{\mathrm{bmi}}$ and $\mathbf{M}$ are known, to use the observed output and hidden-layer activity during training to infer which of the two algorithms was used to train the network. In the case of SL, averaging (8) over noise $\xi$ gives

$$\langle \Delta W_{ij}^{\mathrm{SL}} \rangle = \eta_{\mathrm{SL}} \sum_t [\mathbf{M}\langle \boldsymbol{\epsilon}^t \rangle]_i x_j^t. \tag{10}$$

The expected change in the hidden-layer activity due to learning is then

$$\begin{aligned}
\langle \Delta \mathbf{h}^t \rangle &= \langle \Delta \mathbf{W}^{\mathrm{SL}} \rangle \mathbf{x}^t \\
&= \eta_{\mathrm{SL}} \sum_{t'} (\mathbf{x}^t \cdot \mathbf{x}^{t'}) \mathbf{M} \langle \boldsymbol{\epsilon}^{t'} \rangle.
\end{aligned} \tag{11}$$

In the case of RL, the average of the weight update from (9) is

$$\langle \Delta W_{ij}^{\mathrm{RL}} \rangle = \eta_{\mathrm{RL}} \sum_t [\boldsymbol{\Sigma}(\mathbf{W}^{\mathrm{bmi}})^\top \langle \boldsymbol{\epsilon}^t \rangle]_i x_j^t. \tag{12}$$

The expected change in the hidden-layer activity in this case is then

$$\begin{aligned}
\langle \Delta \mathbf{h}^t \rangle &= \langle \Delta \mathbf{W}^{\mathrm{RL}} \rangle \mathbf{x}^t \\
&= \eta_{\mathrm{RL}} \sum_{t'} (\mathbf{x}^t \cdot \mathbf{x}^{t'}) \boldsymbol{\Sigma}(\mathbf{W}^{\mathrm{bmi}})^\top \langle \boldsymbol{\epsilon}^{t'} \rangle.
\end{aligned} \tag{13}$$

Equations (11) and (13) provide predictions for how the hidden-layer activity is expected to evolve under either SL or RL. Since we do not presume to have knowledge of either the learning rate or the upstream activity $\mathbf{x}^t$, we can concern ourselves only with the direction, but not the magnitude, of $\langle \Delta \mathbf{h}^t \rangle$. In addition, we must make an assumption about correlations between the input patterns. In general, if the input patterns are correlated with one another, we can let $\mathbf{x}^t \cdot \mathbf{x}^{t'} = C(|t - t'|)$, where $C$ is a function with a peak near zero.

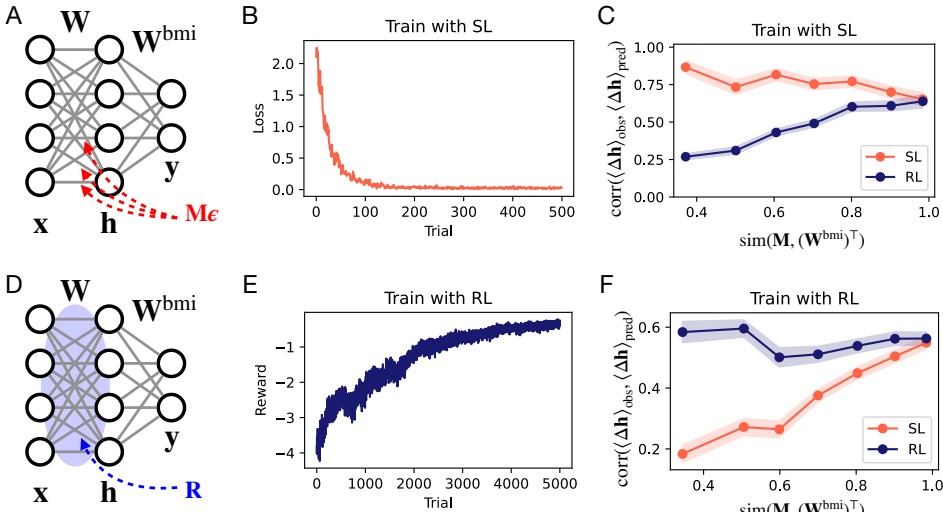

Figure S1: Inference of learning rules from activity in a feedforward model. (A) Hidden-layer weights in a linear network are trained with SL to map random input patterns onto random output patterns. (B) An example showing that the loss successfully decreases during training. (C) Comparison of the observed change in hidden-layer activity during training with the predicted change assuming SL (red) vs. assuming RL (blue). (D-F) Same as top row, but for a network trained with RL.

For the simulations shown in Fig. S1, we have used uncorrelated inputs $\mathbf{x}^t$, so we can assume that $\mathbf{x}^t \cdot \mathbf{x}^{t'} \sim \delta_{tt'}$. With this assumption, Equations (11) and (13) give

$$\langle \Delta \mathbf{h}^t \rangle_{\text{pred}} \propto \begin{cases} \mathbf{M} \langle \boldsymbol{\epsilon}^t \rangle, & \text{SL} \\ \boldsymbol{\Sigma} (\mathbf{W}^{\text{bmi}})^\top \langle \boldsymbol{\epsilon}^t \rangle. & \text{RL} \end{cases} \tag{14}$$

Thus, if we assume that the quantities $\mathbf{M}$, $\boldsymbol{\Sigma}$, and $(\mathbf{W}^{\text{bmi}})^\top$ are known or can be estimated, (14) provides a prediction for how the hidden-layer activity is expected to change due to learning. If this quantity is estimated from data, then the average over noise can be replaced with an empirical average over observed trials: $\langle \ldots \rangle = \frac{1}{N_{\text{trials}}} \sum_{n=1}^{N_{\text{trials}}} (\ldots)$.

If we next suppose that we observe the hidden-layer activity empirically without necessarily knowing the learning rule being used to train the network, then we can define the observed change in hidden-layer activity as the difference between activity observed in early and late trials

$$\langle \Delta \mathbf{h}^t \rangle_{\text{obs}} = \frac{1}{N_{\text{late}}} \sum_{n=1}^{N_{\text{late}}} \mathbf{h}^{n,t} - \frac{1}{N_{\text{early}}} \sum_{n=1}^{N_{\text{early}}} \mathbf{h}^{n,t}, \tag{15}$$

where $\mathbf{h}^{n,t}$ is the activity observed at time $t$ in trial $n$. We can then compare the similarity of $\langle \Delta \mathbf{h}^t \rangle_{\text{obs}}$ with $\langle \Delta \mathbf{h}^t \rangle_{\text{pred}}$ by computing their correlations:

$$\text{corr}(\langle \Delta \mathbf{h}^t \rangle_{\text{obs}}, \langle \Delta \mathbf{h}^t \rangle_{\text{pred}}) = \frac{\sum_i (\langle \Delta h_i \rangle_{\text{obs}} - \overline{\langle \Delta h \rangle_{\text{obs}}})(\langle \Delta h_i \rangle_{\text{pred}} - \overline{\langle \Delta h \rangle_{\text{pred}}})}{\text{std}(\langle \Delta \mathbf{h}^t \rangle_{\text{obs}}) \, \text{std}(\langle \Delta \mathbf{h}^t \rangle_{\text{pred}})}, \tag{16}$$

where $\overline{(\ldots)}$ denotes an average over neurons. In Fig. S1, we show results from a linear feedforward network trained with either SL (Fig. S1A-B) or RL (Fig. S1D-E) to map random input patterns onto random output patterns. Then, using (16), we ask whether the change in the hidden-layer activity during learning in each of these cases is more similar to $\langle \Delta \mathbf{h} \rangle_{\text{pred}}$ predicted assuming SL or RL. The results in Fig. S1C,F show that, whenever $\mathbf{M}$ and $(\mathbf{W}^{\text{bmi}})^\top$ are sufficiently different, this metric is able to correctly identify whether the network was trained with SL (Fig. S1C) or RL (Fig. S1F).

# B Biologically plausible learning rules for recurrent neural networks

In this section we provide derivations of the two learning rules studied in our paper. The RNN update equation is

$$\mathbf{h}^t = \left(1 - \frac{1}{\tau}\right)\mathbf{h}^{t-1} + \frac{1}{\tau}\phi(\mathbf{u}^t) + \boldsymbol{\xi}^t, \tag{17}$$

where $\mathbf{u}^t = \mathbf{W}^{\mathrm{rec}}\mathbf{h}^{t-1} + \mathbf{W}^{\mathrm{fb}}\mathbf{y}^{t-1} + \mathbf{W}^{\mathrm{in}}\mathbf{x}^t$, and $\xi_i^t \sim \mathcal{N}(0, \sigma_{\mathrm{rec}}^2)$ is i.i.d. noise injected into the network. The readout is given by $\mathbf{y} = \mathbf{W}^{\mathrm{bmi}}\mathbf{h}^t$. The goal of both learning rules is to iteratively update the recurrent weights $\mathbf{W}^{\mathrm{rec}}$ such that the readout matches a target function $\mathbf{y}^{*t}$, minimizing the magnitude of the error $\boldsymbol{\epsilon}^t = \mathbf{y}^{*t} - \mathbf{y}^t$. The learning rules that we consider below depend on a multiplicative combination of pre- and postsynaptic activity, as well as a third factor related to error or reward. Evidence for such "three-factor" learning rules has been found in a number of neuroscience experiments [39].

## B.1 Random Feedback Local Online (RFLO) learning

In this section, we briefly recapitulate the derivation of RFLO [5], a supervised learning algorithm for RNNs that uses local weight updates to approximate gradient descent. We then compute the expectation of the weight update by averaging over noise and show that this expected weight update is determined by the credit-assignment matrix $\mathbf{M}$.

The loss function to be minimized is

$$L = \frac{1}{2T}\sum_{t=1}^{T}\sum_{k=1}^{N_y}\left[y_k^{*t} - y_k^t\right]^2. \tag{18}$$

Taking the derivative with respect to the recurrent weights gives

$$\frac{\partial L}{\partial W_{ab}} = -\frac{1}{T}\sum_{t=1}^{T}\sum_{i}\left[(\mathbf{W}^{\mathrm{bmi}})^{\top}\boldsymbol{\epsilon}^t\right]_i\frac{\partial h_i^t}{\partial W_{ab}}. \tag{19}$$

Using (17), we obtain the following recursion relation:

$$\begin{aligned}
\frac{\partial h_i^t}{\partial W_{ab}^{\mathrm{rec}}} = &\left(1 - \frac{1}{\tau}\right)\frac{\partial h_i^{t-1}}{\partial W_{ab}^{\mathrm{rec}}} + \frac{1}{\tau}\delta_{ia}\phi'(u_i^t)h_b^{t-1} \\
&+ \frac{1}{\tau}\phi'\left(u_i^t\right)\left[\sum_j W_{ij}^{\mathrm{rec}}\frac{\partial h_j^{t-1}}{\partial W_{ab}^{\mathrm{rec}}} + \sum_j W_{ij}^{\mathrm{fb}}\frac{\partial y_j^{t-1}}{\partial W_{ab}^{\mathrm{rec}}}\right].
\end{aligned} \tag{20}$$

To obtain a learning rule for $W_{ab}^{\mathrm{rec}}$ that is local, i.e. that depends only on pre- and postsynaptic activity from units $b$ and $a$, respectively, we can discard the second line of this equation and write $\partial h_i^t/\partial W_{ab}^{\mathrm{rec}} \approx \delta_{ia}p_{ab}^t$, where the eligibility trace $p_{ab}^t$ follows the recursion relation

$$p_{ab}^t = \left(1 - \frac{1}{\tau}\right)p_{ab}^{t-1} + \frac{1}{\tau}\phi'\left(u_a^{t-1}\right)h_b^{t-1}, \tag{21}$$

and $p_{ab}^0 = 0$. With this approximation, we arrive at the RFLO learning rule:

$$\Delta W_{ab}^{\mathrm{rec}} = \eta\sum_t[\mathbf{M}\boldsymbol{\epsilon}^t]_a p_{ab}^t. \tag{22}$$

In order to compute the expected change in the RNN flow fields, we wish to compute the expectation of the weight update (22) by averaging over noise. In order to obtain a result for $\Delta\mathbf{F}$ that does not depend explicitly on $\mathbf{W}^{\mathrm{rec}}$, we assume a linear network with $\phi(u) = u$. For convenience, we also switch to the continuous-time limit. In this case, the above eligibility trace in (21) is given by

$$p_{ab} = \int_0^t e^{(s-t)/\tau}h_b^s\frac{ds}{\tau}. \tag{23}$$

Thus, the expected weight update is given by

$$\langle \Delta W_{ij}^{\mathrm{rec}} \rangle = \eta \sum_t \sum_k M_{ik} \int_0^t e^{(s-t)/\tau} \left\langle (y_k^* - y_k^t) h_j^s \right\rangle \frac{ds}{\tau}$$

$$= \eta \sum_t \sum_k M_{ik} \int_0^t e^{(s-t)/\tau} \left( \langle \epsilon_k^t \rangle \langle h_j^s \rangle - \sum_l W_{kl}^{\mathrm{bmi}} Q_{lj}^{t,s} \right) \frac{ds}{\tau}, \tag{24}$$

where we have defined the covariance matrix $\mathbf{Q}^{t,t'} = \langle \mathbf{h}^t (\mathbf{h}^{t'})^\top \rangle - \langle \mathbf{h}^t \rangle \langle \mathbf{h}^{t'} \rangle^\top$. In the case where $\tau$ is unknown (as it may be in experimental data) or sufficiently small relative to the timescale of RNN dynamics, this expression can be simplified by taking the limit $\tau \to 0$, leading to

$$\langle \Delta W_{ij}^{\mathrm{rec}} \rangle = \eta \sum_t \sum_k M_{ik} \left( \langle \epsilon_k^t \rangle \langle h_j^t \rangle - \sum_r W_{kr}^{\mathrm{bmi}} Q_{rj}^{t,t} \right). \tag{25}$$

In the limit where the noise is small, the first term in parentheses will be much larger than the second, which can be dropped to obtain a simplified expression.

## B.2  Reinforcement learning in recurrent neural networks

In this section, we derive a local RNN update rule using policy gradient learning. The resulting "node perturbation" learning algorithm is essentially equivalent to previously proposed RL rules for recurrent circuits from Refs. [28, 10]. We then compute the expectation of the weight update (34) by averaging over noise and show that this expected weight update is determined by the decoder $\mathbf{W}^{\mathrm{bmi}}$, following the unbiased gradient direction.

In policy gradient learning, a policy $\pi(\mathrm{action}|\mathrm{state})$ is optimized with respect to its parameters in order to maximize a scalar performance measure $R^t$. In our case, we interpret $\mathbf{h}^t$ as the action, $\mathbf{h}^{t-1}$ as the state, and $\mathbf{W}^{\mathrm{rec}}$ as the parameters to be optimized. We take the policy to be

$$\pi(\mathbf{h}^t | \mathbf{h}^{t-1}, \mathbf{W}^{\mathrm{rec}}) \sim \mathcal{N}(\mathbf{h}^t | \boldsymbol{\mu}^t, \sigma_{\mathrm{rec}}^2 \mathbf{I}), \tag{26}$$

where

$$\boldsymbol{\mu}(t, \mathbf{W}^{\mathrm{rec}}) = \left( 1 - \frac{1}{\tau} \right) \mathbf{h}^{t-1} + \frac{1}{\tau} \phi(\mathbf{u}^t) \tag{27}$$

is the deterministic part of the update equation (17). The policy gradient theorem allows us to update the policy parameters in a way that ensures improvement of the objective $R^t$. The REINFORCE algorithm [8] is based on the policy gradient theorem and updates the parameters $\mathbf{W}^{\mathrm{rec}}$ according to

$$\Delta \mathbf{W}^{\mathrm{rec}} \propto \left( R^t - \bar{R}^t \right) \nabla \ln \pi(\mathbf{h}^t | \mathbf{h}^{t-1}, \mathbf{W}^{\mathrm{rec}}), \tag{28}$$

where the gradient is with respect to $\mathbf{W}^{\mathrm{rec}}$. The reward baseline $\bar{R}^t$, to be defined below, is not required for policy gradient learning but can decrease the variance of the updates [11].

The gradient can be computed as follows:

$$\frac{\partial}{\partial W_{ab}^{\mathrm{rec}}} \ln \pi(\mathbf{h}^t | \mathbf{h}^{t-1}, \mathbf{W}^{\mathrm{rec}}) = -\frac{1}{2\sigma_{\mathrm{rec}}^2} \frac{\partial}{\partial W_{ab}^{\mathrm{rec}}} \left( \mathbf{h} - \boldsymbol{\mu}(t, \mathbf{W}^{\mathrm{rec}}) \right)^2$$

$$= \frac{1}{\sigma_{\mathrm{rec}}^2} \sum_i [h_i - \mu_i(t, \mathbf{W}^{\mathrm{rec}})] \frac{\partial}{\partial W_{ab}^{\mathrm{rec}}} \mu_i(t, \mathbf{W}^{\mathrm{rec}})$$

$$= \frac{1}{\sigma_{\mathrm{rec}}^2} \sum_i \xi_i^t \frac{\partial}{\partial W_{ab}^{\mathrm{rec}}} \mu_i(t, \mathbf{W}^{\mathrm{rec}})$$

$$= \frac{1}{\tau \sigma_{\mathrm{rec}}^2} \xi_a^t \phi'\left( u_a^t \right) h_b^{t-1}. \tag{29}$$

In order to address temporally delayed credit assignment, we can additionally incorporate an eligibility trace in the gradient appearing in (28), replacing $\nabla \ln \pi \longrightarrow \overline{\nabla \ln \pi}$, where the bar denotes low-pass filtering [11]. This allows credit for rewards at time $t$ to be assigned to the RNN activity at earlier time steps. This leads to the following update rule:

$$\Delta W_{ab}^{\mathrm{rec}} = \eta(R^t - \bar{R}^t) q_{ab}^t, \tag{30}$$

where the eligibility trace is given by

$$q_{ab}^t = \left(1 - \frac{1}{\tau_e}\right) q_{ab}^{t-1} + \frac{1}{\tau_e} \xi_a^t \phi'(u_a^t) h_b^{t-1}, \tag{31}$$

with $q_{ab}^0 = 0$. In our simulations, we set the timescale for the eligibility trace to be equal to the network time constant, i.e. $\tau_e = \tau$.

In order to compute the expected change $\Delta \mathbf{F}^{\text{pred}}$ in the RNN flow fields, we next wish to compute the expectation of the weight update (30) by averaging over noise. As in the SL case from the previous section, we assume that the RNN is linear and switch to the continuous-time limit. In this case, the eligibility trace is given by

$$q_{ij}^t = \int_0^t e^{(s-t)/\tau_e} \xi_i^s h_j^s \frac{ds}{\tau_e}.$$

The reward $R^t$ itself is given by

$$R^t = -|\boldsymbol{\epsilon}^t|^2 = -|\mathbf{y}^{*t}|^2 - |\mathbf{y}^t|^2 + 2\mathbf{y}^{*t} \cdot \mathbf{y}^t. \tag{32}$$

We assume that the expected reward $\bar{R}^t$ is independent of the noise $\xi$, so it doesn't contribute to $\langle \Delta W^{\text{rec}} \rangle$. Then we can use Wick's theorem to calculate the expected weight update:

$$\langle \Delta W_{ij}^{\text{rec}} \rangle = \eta \sum_t \sum_k \int_0^t e^{(s-t)/\tau_e} \left\langle (2y_k^* y_k^t - y_k^t y_k^t) \xi_i^s h_j^s \right\rangle \frac{ds}{\tau_e}$$

$$= \eta \sum_t \sum_k \int_0^t e^{(s-t)/\tau_e} \left( 2\langle y_k^* h_j^s \rangle - 2\langle y_k^t h_j^s \rangle \right) \langle \xi_i^s y_k^t \rangle \frac{ds}{\tau_e} \tag{33}$$

$$= 2\eta \sum_t \sum_{k,l} W_{kl}^{\text{bmi}} \int_0^t e^{(s-t)/\tau_e} \left( \langle y_k^* h_j^s \rangle - \langle y_k^t h_j^s \rangle \right) \langle \xi_i^s h_l^t \rangle \frac{ds}{\tau_e}.$$

For the linear network we are considering, we have $\langle \xi_i^s h_l^t \rangle = \sum_m \Sigma_{im} \left( e^{\mathbf{W}(t-s)} \right)_{lm}$, where $\boldsymbol{\xi}^s \sim \mathcal{N}(0, \boldsymbol{\Sigma})$ and $\mathbf{W} = -\mathbb{I} + \mathbf{W}^{\text{rec}} + \mathbf{W}^{\text{fb}} \mathbf{W}^{\text{bmi}}$. Then

$$\langle \Delta W_{ij}^{\text{rec}} \rangle = 2\eta \sum_t \sum_{k,l,m} W_{kl}^{\text{bmi}} \Sigma_{im} \int_0^t e^{\frac{s-t}{\tau_e}} \left( e^{\mathbf{W}(t-s)} \right)_{lm} \left( \langle \epsilon_k^t \rangle \langle h_j^s \rangle - \sum_r W_{kr}^{\text{bmi}} Q_{rj}^{t,s} \right) \frac{ds}{\tau_e}. \tag{34}$$

In the case where $\tau_e$ is unknown (as it may be in experimental data) or sufficiently small, this expression can be simplified by taking the limit $\tau_e \to 0$, leading to

$$\langle \Delta W_{ij}^{\text{rec}} \rangle = 2\eta \sum_t \sum_{k,l} W_{kl}^{\text{bmi}} \Sigma_{il} \left( \langle \epsilon_k^t \rangle \langle h_j^t \rangle - \sum_r W_{kr}^{\text{bmi}} Q_{rj}^{t,t} \right). \tag{35}$$

As in the SL case, in the limit where the noise is small, the first term in parentheses will be much larger than the second, which can be dropped to obtain a simplified expression.

## C  Simulations

### C.1  Experimental details

The code used to run these simulations can be found at `www.github.com/jacobfulano/learning-rules-with-bmi`

As stated in the main text, we first pretrain the recurrent weights $\mathbf{W}^{\text{rec}}$ of an RNN with fixed, random readout weights $\mathbf{W}^{\text{bmi0}}$ to perform a center-out cursor-control task, in which a cursor must be moved to one of four target locations specified by the input to the RNN. We then change the BMI decoder to $\mathbf{W}^{\text{bmi1}}$, make identical copies of the network, and train one with SL and the other with RL. Multiple seeds ($n = 4$) were selected for $\mathbf{W}^{\text{bmi1}}$ and $\mathbf{M}$ and applied to network copies in the training phase, and correlation metric results $\text{Corr}(\Delta \mathbf{F}^{\text{obs}}, \Delta \mathbf{F}^{\text{pred}})$ were averaged over these seeds. Error bars are S.E.M. Unless stated otherwise, all simulations involved pretraining.

Simulations for Figures 2 used 4 input dimensions, $N = 50$ recurrent units, and 2 output dimensions, with a trial duration of 20 timesteps. The scale for the variance of the noise injected at each layer was $\sigma_{\text{in}}^2 = 0, \sigma_{\text{rec}}^2 = 0.25, \sigma_{\text{bmi}}^2 = 0.01$ for input, recurrent, and output units respectively. Constant learning rate for recurrent weights was $\eta^{\text{rec}} = 0.1$, and the RNN time constant was set to $\tau = 10$. Recurrent weights $\mathbf{W}^{\text{rec}}$ were initialized with $\mathbf{W}^{\text{rec}} \sim \mathcal{N}(0, g^2/\sqrt{N})$ where $g = 1.5$. Input weights $\mathbf{W}^{\text{in}}$ and decoder weights $\mathbf{W}^{\text{bmi}}$ were initialized randomly and uniformly over $[-2, 2]$ and $[-2/\sqrt{N}, 2/\sqrt{N}]$ respectively. For all networks, the activation function was $\phi(\cdot) = \tanh(\cdot)$. Pretraining was run for 2,500 trials using SL, and training was run for 1,500 trials for SL and 15,000 trials for RL. Alignment between $\mathbf{M}$ and $\mathbf{W}^{\text{bmi0}}$ was fixed at 0.5. Input signals for each target consisted of a step function that was 1 for 20% of the trial duration (i.e. 4 timesteps) and 0 for the remainder. Input to the network at each timestep was therefore a 4 dimensional vector with one entry equal to 1 and other entries equal to zero. For both SL and RL algorithms, weights were updated at the end of each trial (i.e. "offline"). For RL simulations using (3), a separate reward baseline $\bar{R}^t$ was kept for each target.

In order to control alignment $\alpha$ between decoder weights $(\mathbf{W}^{\text{bmi}})^\top$ and matrices $\mathbf{M}$, we generated a matrix $\mathbf{M}$ by randomly changing a subset of matrix entries from $(\mathbf{W}^{\text{bmi}})^\top$ such that $\text{sim}(\mathbf{M}, (\mathbf{W}^{\text{bmi}})^\top) = \alpha$. Networks trained with SL were able to consistently learn the task with four targets for $\alpha > 0.3$. Throughout this study, analyses were only performed on networks that successfully learned the center-out reach task.

When calculating the flow field metric in (7), "early" (before learning) and "late" (after learning) blocks consisted of 500 trials. Activity during learning was split into training trials and test trials; predictions of the (direction of) weight change $\Delta \mathbf{W}^{\text{pred}}|_{SL}$ and $\Delta \mathbf{W}^{\text{pred}}|_{RL}$ were constructed using activity $\mathbf{h}^{n,t}$ and error $\boldsymbol{\epsilon}^{n,t}$ from the training trails, and the full metric $\text{Corr}(\Delta \mathbf{F}^{\text{obs}}(\mathbf{h}), \Delta \mathbf{F}^{\text{pred}}(\mathbf{h}))$ was evaluated on activity $\mathbf{h}^{n,t}$ from the test trials.

Fig. S2 shows that the flow field correlation metric successfully distinguishes the learning rules across hyperparameters, including recurrent noise $\sigma_{\text{rec}}^2$ and number of recurrent units, for both SL and RL. For networks with a large number of recurrent units, we found that the RL node perturbation algorithm was more effective when the noise was low-dimensional. In Fig. S2D, therefore, the recurrent noise is 50-dimensional, and isotropic within those dimensions.

For Fig. 3A, weight mirroring [6] was used as a convenient way to update $\mathbf{M}$ while also learning recurrent weights $\mathbf{W}^{\text{rec}}$ with SL. Network parameters were the same as Fig. 2, except that $\eta^{\text{rec}}$ was lowered to 0.05 and the weight mirroring learning rate was $\eta^{\text{WM}} = 0.001$. The weight mirroring algorithm applied to our context simply correlates presynaptic recurrent noise $\xi_j \sim \mathcal{N}(0, \sigma_{\text{rec}}^2)$ with postsynaptic activity $y_i = W_{ij}^{\text{bmi}} \xi_j$ and then updates $\mathbf{M}$ with a update rule $\Delta M_{ji} = \eta^{\text{WM}} \xi_j y_i$. On average, this pushes $\mathbf{M}$ in the direction of $\mathbb{E}[\xi_j y_i] = \sigma_{\text{rec}}^2 W_{ji}^{\text{bmi}}$, i.e. the transpose of the decoder weights $\mathbf{W}^{\text{bmi}}$.

For Fig. 3C-E, we included driving feedback weights $\mathbf{W}^{\text{fb}} = \gamma \mathbf{M}$ and varied the strength of the driving feedback weights by a scalar factor $\gamma$ between 0.5 and 5. For these simulations, $\mathbf{M}$ was kept fixed at $\text{sim}(\mathbf{M}, (\mathbf{W}^{\text{bmi}})^\top) = 0.5$. Training ran for 5,000 trials for networks using SL, and 10,000 trials for networks using RL. Network parameters were the same as Fig. 2, except with $\eta^{\text{rec}} = 0.1$ and $\sigma_{\text{rec}}^2 = 0.1$.

For Fig. 4, the stimulus signal was equal to 1 for the full trial, and the feedback weights were set to $\mathbf{W}^{\text{fb}} = \gamma \mathbf{M}$ with $\gamma = 5$. Pretraining lasted for 2,500 trials, while training was set to 1,000 trials and the overlap between $\mathbf{M}$ and $\mathbf{W}^{\text{bmi0}}$ was set to 0.5. Network parameters were the same as Fig. 2, except with $\eta^{\text{rec}} = 1$, and $\sigma_{\text{rec}}^2 = 0.2$.

For Fig. 5, we trained networks to learn a cursor-control task with non-isotropic noise via either SL or RL. In the SL case, we chose a biased credit mapping $\mathbf{M}$ that has partial overlap with the new decoder, with $\text{sim}(\mathbf{M}, (\mathbf{W}^{\text{bmi1}})^\top) = 0.6$. Network parameters were the same as Fig. 2, except with $\eta^{\text{rec}} = 0.2$. Training was run for 1,000 trials. The recurrent noise covariance was chosen to be $d$-dimensional, with $\text{rank}(\boldsymbol{\Sigma}) = d$, and isotropic within those dimensions. Simulations were run for $d = 5, 10, 25, 50$ with three seeds for each dimension. The first two of these dimensions were selected to lie in the subspace spanned by $\mathbf{M}$ for the SL-trained RNNs or by $\mathbf{W}^{\text{bmi1}}$ for the RL-trained RNNs via QR decomposition. Other components of $\boldsymbol{\Sigma}$ were added in random dimensions orthogonal to this subspace and to one another.

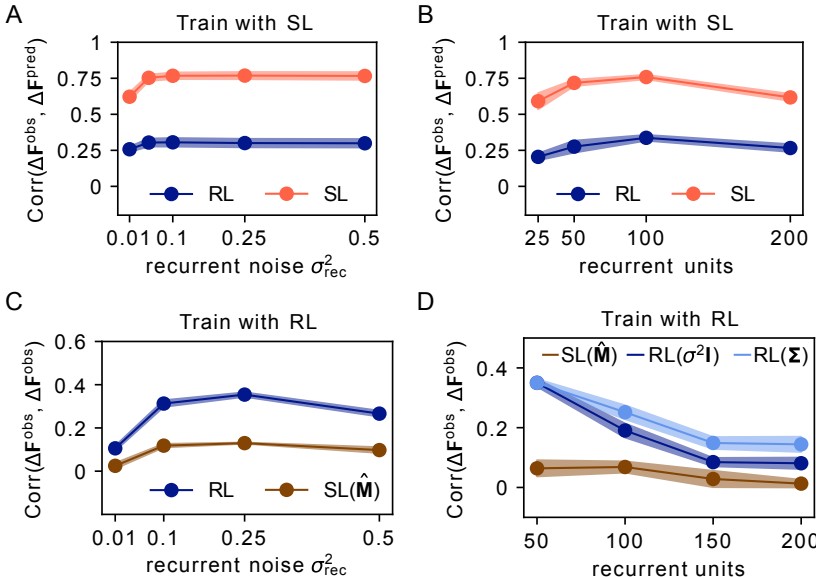

Figure S2: Learning rules are distinguishable across hyperparameter choices. (A) Varying recurrent noise while training with SL for networks with 50 recurrent units. (B) Varying the number of recurrent units while training with SL. Both (A) and (B) were obtained using $\mathrm{sim}(\mathbf{M}, (\mathbf{W}^{\mathrm{bmi1}})^\top) = 0.5$, with other hyperparameters the same as in Fig. 2. (C) Varying recurrent noise while training with RL for networks with 50 recurrent units. (D) Varying the number of recurrent units while training with RL. Noise dimension was set to 50, and correlation was calculated for true, low-dimensional noise covariance (light blue), or naive, full dimension estimate of the noise covariance (dark blue). For both (C) and (D), $\hat{\mathbf{M}}$ was randomly sampled such that $\mathrm{sim}(\hat{\mathbf{M}}, (\mathbf{W}^{\mathrm{bmi1}})^\top) = 0.5$; other hyperparameters are the same as in Fig. 2.

In Fig. S1, the network size was 20-20-2, and the task was to map $T = 5$ random binary input patterns onto random output targets. For SL, the parameters were $\eta_{\mathrm{SL}} = 0.001$, $\sigma = 0.1$, $N_{\mathrm{trials}} = 500$, and $N_{\mathrm{early}} = N_{\mathrm{late}} = 10$. For RL, the parameters were $\eta_{\mathrm{SL}} = 0.003$, $\sigma = 0.1$, $N_{\mathrm{trials}} = 5000$, and $N_{\mathrm{early}} = N_{\mathrm{late}} = 100$. In both cases, the results shown in Fig. S1C,F were computed by averaging over 100 different networks for each condition.

## C.2 Alternative supervised learning rules: "Biased" Backpropagation Through Time (BPTT)

As described in the main text, RFLO [5] is an approximate gradient-based algorithm with $\Delta \mathbf{W}^{\mathrm{rec}} \approx -\partial L/\partial \mathbf{W}^{\mathrm{rec}}$. The main results of this study depend on the key idea that the matrix $\mathbf{M}$ used in the (supervised) learning rule is not identical to the transpose of the decoder weights $(\mathbf{W}^{\mathrm{bmi}})^\top$. This idea can be applied to BPTT, leading to a learning rule we call "biased" BPTT. The standard BPTT update is

$$\frac{\partial L}{\partial W_{ab}^{\mathrm{rec}}} = -\frac{1}{\tau T} \sum_t z_a^t \phi'(u_a^t) h_b^{t-1} \tag{36}$$

$$z_i^t = \sum_j W_{ji}^{\mathrm{bmi}} \epsilon_j^t + \left(1 - \frac{1}{\tau}\right) z_i^{t+1} + \frac{1}{\tau} \sum_j \phi'(u_j^{t+1}) W_{ji}^{\mathrm{rec}} z_j^{t+1} \tag{37}$$

with $u_i^t = \sum_k W_{ik}^{\mathrm{rec}} h_k^{t-1} + \sum_k W_{ik}^{\mathrm{in}} x_k^t + \sum_k W_{ik}^{\mathrm{fb}} y_k^t$, as in the main text. For a careful comparison of BPTT, RFLO, and other related gradient-based algorithms for training RNNs, see Refs. [5, 40].

A *biased* BPTT learning rule according to our framework would simply replace the $(\mathbf{W}^{\mathrm{bmi}})^\top \epsilon^t$ term with $\mathbf{M}\epsilon^t$:

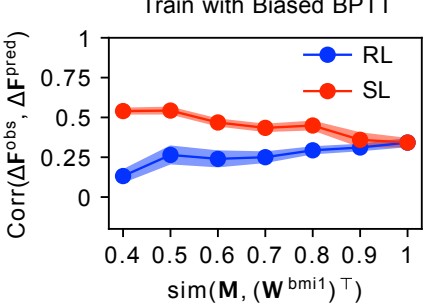

Figure S3: Correlation metric generalizes to other SL algorithms. Networks are trained with a biased form of BPTT, where $(\mathbf{W}^{\mathrm{bmi1}})^\top$ is replaced by $\mathbf{M}$ in the weight update rule.

$$z_i^t = \sum_j M_{ij}\epsilon_j^t + \left(1 - \frac{1}{\tau}\right)z_i^{t+1} + \frac{1}{\tau}\sum_j \phi'(u_j^{t+1})W_{ji}^{\mathrm{rec}}z_j^{t+1} \qquad (38)$$

We show in Fig. S3 that our results hold when using a biased BPTT learning rule instead of the supervised RFLO learning rule. Network parameters and simulation details were the same as Fig. 2, except that a biased BPTT learning rule was used. For lower similarity between $\mathbf{M}$ and $(\mathbf{W}^{\mathrm{bmi}})^\top$, the correlation metric is able to correctly identify bias in the change in flow field. Compared with the results shown in Fig. 2, the correlation metric values are somewhat lower for biased BPTT. This is likely because additional nonlocal recurrent terms are contained in the true weight update that are not accounted for in the expression for $\Delta\mathbf{F}_{\mathrm{SL}}^{\mathrm{pred}}$ that uses $\Delta\mathbf{W}^{\mathrm{pred}}|_{\mathrm{SL}}$ from (5).

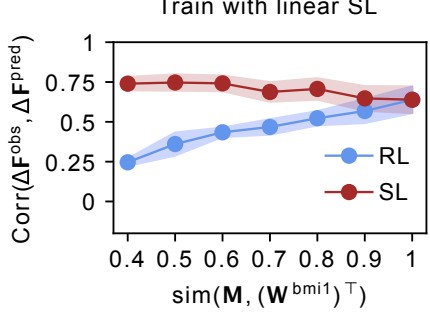

Figure S4: Correlation metric generalizes to linear networks.

## C.3 Linear RNNs

We show that the RNN nonlinearity $\phi(\cdot) = \tanh(\cdot)$ is not a crucial architectural choice for our main conclusions. In Fig. S4 we apply the same approach (and hyperparameters) as shown in Fig. 2C to linear RNNs and find similar results.

## C.4 Velocity-based cursor control

In the main text, we modeled a task in which a BMI readout maps neural activity directly onto cursor position. While this is a conceptually simple way to illustrate our main ideas, actual BMI experiments more commonly use the readout of neural activity to control cursor *velocity* rather than position (e.g. Refs. [12, 15, 16, 17]). In this section we show that our main results can also be obtained for this case.

Let the RNN readout $\mathbf{y}^t = \mathbf{W}^{\mathrm{bmi}}\mathbf{h}^t$ correspond to cursor velocity rather than to cursor position, and let the cursor position be given by $\mathbf{r}^t = (1 - 1/\tau_r)\mathbf{r}^{t-1} + \mathbf{y}^t/\tau_r$. For simplicity, we take $\tau_r = \tau$ in

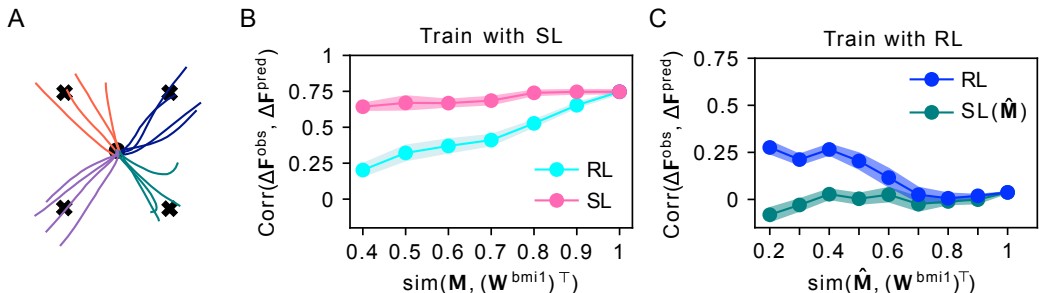

Figure S5: SL and RL are distinguishable with velocity-based cursor control. (A) Example trajectories from a trained RNN in which BMI readout weights map neural activity to cursor velocity. (B) $\mathrm{Corr}(\Delta\mathbf{F}^{\mathrm{obs}}, \Delta\mathbf{F}^{\mathrm{pred}})$ for RNNs trained with SL to control cursor velocity. (C) $\mathrm{Corr}(\Delta\mathbf{F}^{\mathrm{obs}}, \Delta\mathbf{F}^{\mathrm{pred}})$ for RNNs trained with RL to control cursor velocity.

our simulations. Let the target velocity at time $t$ be given by $\mathbf{y}^{*t} = \mathbf{r}^* - \mathbf{r}^t$, where $\mathbf{r}^*$ is the target position, and the error be given by $\boldsymbol{\epsilon}^t = \mathbf{y}^{*t} - \mathbf{y}^t$.

Using velocity-based cursor control leads to smoother cursor trajectories, as shown in Fig. S5A. Figures S5B-C show that the learning rules can be correctly identified in RNNs that are trained under these assumptions using either SL or RL.

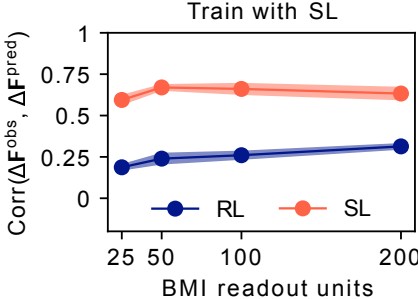

Figure S6: Correlation metric applied to networks with 200 recurrent units. BMI decoders only read out activity from a subset of neurons in the recurrent population.

### C.5   Learning with a BMI that samples from a subset of the neural population

The simulations in this study have assumed that the decoder $\mathbf{W}^{\mathrm{bmi}}$ reads out the neural activity from all the neurons in the RNN. While this assumption is not realistic with respect to neuroscience experiments, we show here that it does not affect our conclusions.

We ran simulations for RNNs with 200 recurrent units using SL and varied the number of units read out by the decoder between 25 and 200, setting the decoder weights of all non-readout units to zero. After pretraining the RNN, the decoder $\mathbf{W}^{\mathrm{bmi1}}$ was randomly selected such that $\mathrm{sim}(\mathbf{W}^{\mathrm{bmi0}}, \mathbf{W}^{\mathrm{bmi1}}) = 0.5$, while reading out from the same units that were read out from during pretraining. The credit assignment mapping $\mathbf{M}$ was randomly chosen such that $\mathrm{sim}(\mathbf{M}, (\mathbf{W}^{\mathrm{bmi1}})^{\top}) = 0.5$, and was not necessarily restricted to the same subset of readout units. This was repeated with different numbers of readout units, each across 5 random seeds. Network and training hyperparameters were otherwise the same as in Figures 2 and S2. Fig. S6 shows that, in this more realistic scenario where a BMI only decodes a subset of the neurons in the neural population, our correlation metric continues to distinguish between the SL and RL training algorithms.

### C.6   Weight updates are distinct for biased SL and RL

In order to build an intuition for how the SL and RL rules affect weight updates, we analyzed these updates directly (rather than the changes in flow fields, which was the focus in the main text).

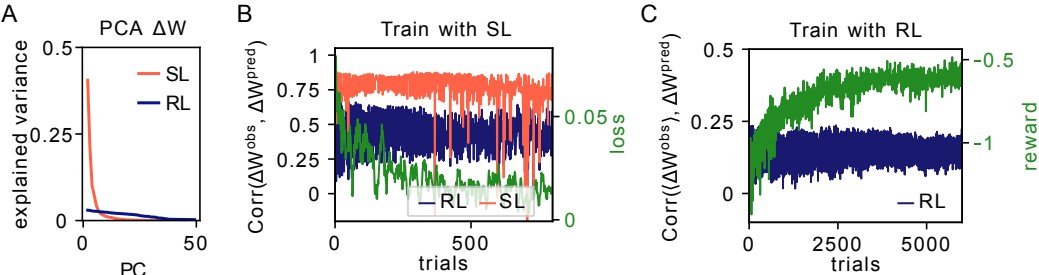

Figure S7: Observed weight updates follow the predicted update direction. (A) PCA on the weight changes for a network trained with SL (red) and for a network trained with RL (blue). (B) In a network trained with SL, the observed weight updates from individual trials are correlated with $\Delta\mathbf{W}^{\mathrm{pred}}$ predicted by SL (using matrix $\mathbf{M}$). (C) For a network trained with RL, the observed weight updates averaged over trials are correlated with $\Delta\mathbf{W}^{\mathrm{pred}}$ predicted by RL (using matrix $\mathbf{W}^{\mathrm{bmi}}$).

In the SL case we use (5):

$$\Delta W_{ij}^{\mathrm{pred}}|_{\mathrm{SL}} = \sum_t^T \sum_k M_{ik}\epsilon_k^t h_j^t.$$

In the RL case we use (6):

$$\Delta W_{ij}^{\mathrm{pred}}|_{\mathrm{RL}} = \sum_t^T \sum_k W_{kl}^{\mathrm{bmi}}\Sigma_{il}\epsilon_k^t h_j^t,$$

with isotropic noise $\mathbf{\Sigma} = \sigma_{\mathrm{rec}}^2 \mathbf{I}$.

Fig. S7B shows that, for an RNN trained with SL, the observed weight updates $\Delta\mathbf{W}^{\mathrm{obs}}$ for each trial are highly correlated with the predicted weight updates $\Delta\mathbf{W}^{\mathrm{pred}}|_{\mathrm{SL}}$. This is particularly pronounced early in learning, as the loss is still decreasing. This analysis is applied to a network trained with SL with $\mathrm{sim}(\mathbf{M}, (\mathbf{W}^{\mathrm{bmi}})^\top) = 0.6$, and is one way of building intuition for the results in Fig. 2C and the FFCC metric $\mathrm{Corr}(\Delta\mathbf{F}^{\mathrm{obs}}, \Delta\mathbf{F}^{\mathrm{pred}})$.

The RL updates look quite different. Principal component analysis on the weight changes for a network trained with RL shows that weight updates are much more spread out across PCs (Fig. S7A), which also indicates that individual weight updates don't follow one direction in the loss landscape. In Fig. S7C, however, the *average* of the observed weight updates over trials during learning $\langle\Delta\mathbf{W}^{\mathrm{obs}}\rangle$ is correlated with $\Delta\mathbf{W}^{\mathrm{pred}}|_{\mathrm{RL}}$ predicted after each trial throughout learning.

### C.7 Statistical significance of simulations

To confirm the statistical significance of our results and their dependence on the level of noise in the RNN, we repeat the simulations from Figure 2 with different levels of noise in Figure S8 and show that predictions of the two learning rules are statistically distinct when the alignment between the BMI decoder and the credit assignment weights is sufficiently small.

### C.8 Change of neural activity manifold with training

In order to predict the flow field change for a point $\mathbf{h}$ in neural activity space, the direction of weight change is predicted via (5) and (6) by sampling activity from trials during training. In order to verify that there is significant overlap of the distributions from which $\mathbf{h}$ is sampled at different points during training and retraining, we compared the covariance matrices of the RNN activity (Figure S9A) from early trials vs. later trials with the same decoder (Figure S9B, green curve) or trials after retraining on a new decoder (Figure S9B, magenta curve). These results show that, although the alignment between the neural activity manifolds before vs. after retraining with a new decoder decreases as the old and new decoders become less similar, a significant degree of overlap remains even when the decoders are highly dissimilar.

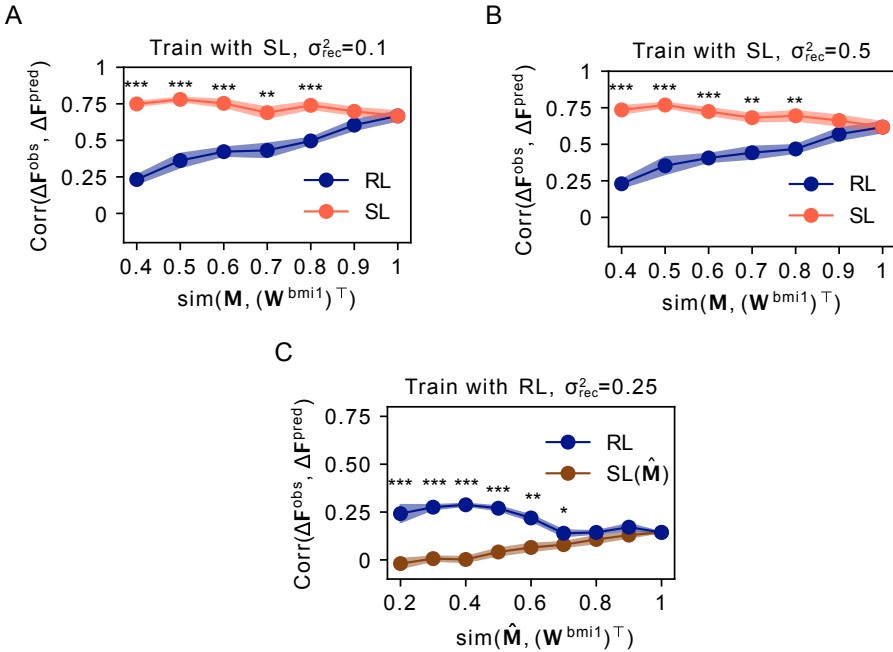

Figure S8: Statistical significance of the correlation metric. (A) The std of the recurrent noise was set to 0.1, with the simulations being otherwise identical to the simulations in Fig. 2C. (B) Same as in (A), but with the recurrent noise set to 0.5. (C) RL simulation data from Fig. 2F. Two sample t-test: (*) indicates $p < 0.05$, (**) indicates $p < 0.01$, and (***) indicates $p < 0.001$.

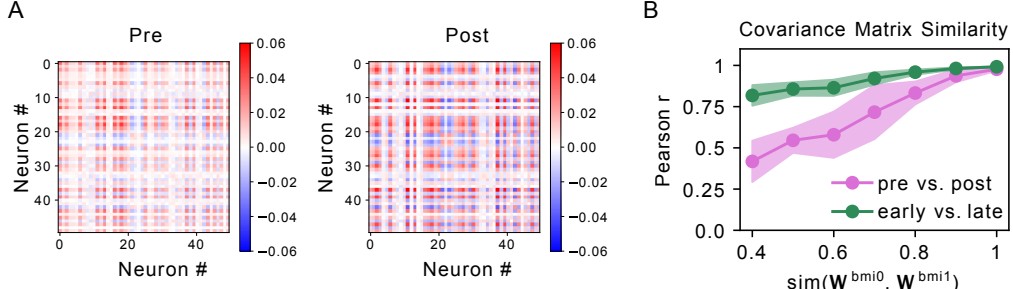

Figure S9: Change of neural activity manifold with training. (A) Example covariance matrices for activity before ("pre") and after ("post") learning a new decoder via SL. For these examples, $\text{sim}(\mathbf{W}^{\text{bmi0}}, \mathbf{W}^{\text{bmi1}}) = 0.8$ (B) Pearson $r$ for pairs of covariance matrices as a function of the alignment between $\mathbf{W}^{\text{bmi0}}$ and $\mathbf{W}^{\text{bmi1}}$. Magenta line shows the correlation of covariance matrices for activity before ("pre") and after ("post") learning a new decoder via SL. Green line shows the Pearson correlation of covariance matrix pairs for activity in the first half ("early") and second half ("late") of learning a new decoder via SL (n=3 seeds; error bars show standard deviation).