# OpenReview forum: "Distinguishing Learning Rules with Brain Machine Interfaces"
_NeurIPS.cc/2022/Conference — NeurIPS 2022 Accept_

### Official Review · Reviewer_GGdc · 2022-06-23

**Rating:** 7
**Confidence:** 4
**Soundness:** 2 fair
**Presentation:** 3 good
**Contribution:** 2 fair

**Summary:**

The authors provide a framework to distinguish biased vs unbiased learning rules by observing only the unit activity and demonstrate the usability of the framework for recurrent neural networks. The authors demonstrate that observing the changes in flow field is indicative of the learning rule that leads to synaptic weight changes and can thus be used to distringuish if the learning rule was biased or unbiased wrt the true gradient signal. Subsequently, the results show that the framework can distinguish the two classes of learning rules when there is sufficient bias in one of them, i.e. the similarity between the credit assignment mapping and the true readout weights is low. Furthermore, they add more factors, namely the feedback weights, anisotropic noise as well as a changing credit assignment mapping matrix and show that the proposed framework can identify bias in the learning rule. Given the results, it would be interesting to see how this framework extends to realistic setting with neural data from BMI experiments.

**Questions:**

1. See weakeness point 1
2. I am unsure why the RL line in Fig 2F fluctuates with changing $\hat{M}$. In my understanding $\hat{M}$ is only used by the SL predictions and therefore I can understand the different values of the SL line.
3. While computing $\Delta F$, don't you need the same $h$ for both $F_late$ and $F_early$? Is this achieved in the current setting by setting the t to be the same for early and late stages? How can this be achieved in the realistic scenario?
4. In the RHS of Eq 4, I am assuming by $h$, you mean $h^{t}$. Is that correct?
5. The relation between $\Delta F$ and $\Delta W^{rec}$ only holds when it is assumed that $W^{in}$ or $W^{fb}$ is unchanged. Is that a reasonable assumption for realistic BMI settings? If yes, kindly add citations. If not, I'd suggest adding that to the limitations section of your work.
6. Furthermore, the relation between $\Delta F$ and $\Delta W^{rec}$ assumes same values of $x,y$. Assuming that the early and late stages of your estimation deals with similar inputs (hence same $x$), the output of the system should surely change given that there is some learning. So isn't it unreasonable to assume that $y$ would be same in both early vs late stages?
7. It is a bit hard to grasp the correlation values in the plots. Although the relative values are higher for the correct learning rule, the absolute values are not very informative. Is it possible to add some sort of "noise ceiling" to indicate what could be the best achievable correlation value from theory such that it is possible to compare the efficacy of the framework in correctly identifying the ground truth learning rule?
8. The framework seems to be able to distinguish between two learning rules when one has significant bias and the other does not. Can it also provide an indication as to what amount of bias is the "correct" amount of bias in the weight updates wrt the gradient? Given the current results, it seems unlikely but I am curious to know if you have further insights into this.

**Limitations:**

Overall, I think the authors do a commendable job in presenting their framework. If the issues highlighted above can be addressed, I feel this framework could have significant impact in evaluating the plausibility of different learning rules in the brain. Currently, this work has certain limitations:
1. A major limitation is the presentation of the flow field change as a way to understand the change in synaptic weights and how the correlation metric is a good summary metric that is indicative of this change. Specifically, I feel it is not immediately clear how the absolute values can be used to identify the correct learning rule or indicate if a learning rule is not plausible.
2. If the authors choose to present their framework as being a good way to compare the biological realism of learning rules, I feel they would need to add more learning rules and show how the metric can either rank the "closeness" to the true learning rule or select the correct learning rule from several candidate learning rules. Although this could involve significant experimentation, I feel this would be helpful in establishing the credibility of the framework as a way to order the different proposals currently acknowledged in the field.
3. The framework lacks analysis of high variance learning rules, which would be crucial in establishing the tradeoff/impact of bias and variance in bio-plausible learning rules and thereby guide the field into proposing learning rules with specific constraints in mind.
4. It would be great if the authors could add a potential simulation of how the framework would perform in realistic BMI settings where the recording SNR could be a major bottleneck. However, I understand that this point is more a future work and might not fit in the current scope, if the authors choose to present it as such.

**Strengths And Weaknesses:**

Strengths:
1. The idea of using flow field changes to infer changes in synaptic weight is neat and seems promising. The authors explain the key learning rules that they explore as well as the flow field well.
2. Overall, the paper is well-motivated and well-presented. It is easy to understand the main goal and the general approach of the paper.
3. The authors systematically add various aspects and complexities of the system, which provides an insight into how the different components interact with the proposed framework.
4. The results are encouraging enough for the community to extend this work and either use it to analyze other learning rules or use it for BMI experiments.

Weakness:
1. I felt the later half of the results section a bit hard to read or follow. Specifically, it was not very clear how the solution to the autoregression problem in h translates to being the mapping from h to F(h). If $h^{t+1} = A h^{t}$, then shouldn't $F(h^{t}) = h^{t+1}-h^{t} = (A-I)h^{t}$, i.e. shouldn't the mapping from $h$ to $F(h)$ be $(A-I)$ instead of $A$ ?
2. The correlation metric values are not setting invariant, i.e. the absolute values are not necessarily indicative of the correct learning rule. For instance, the correlation values for the RL-predicted flow changes are similar at low values of $sim(M,(W^{bmi1})^T)$ and $sim(\hat{M},(W^{bmi1})^T)$ in Fig 2C and 2F respectively, although the ground truth learning rule is different. Therefore, it is not very clear how to use the value in realistic experimental settings to identify the correct learning rule. In my understanding, the metric can be used to compare between candidate learning rules but given that correlation values of the ground truth can also be as low as 0.25, it is unclear how reliable it would be for other learning settings.
3. This work provides a comparison of two learning rules - biased Supervised Learning and unbiased Reiforcement Learning. The proposed framework is able to distinguish (barring some specific cases) between the two rules. However, it is unclear how noise in the learning rule would impact the framework. Specifically, the authors used a noise-averaged version of the RL weight update as the learning rule. But in realistic scenario, weight updates could be characterized by noise around the true gradient direction (high variance update as noted in Fig 1). It is unclear if the framework is relevant for distinguishing such learning rules.
4. The plots are missing statsitical significance test to infer if the metric is statistically significant in the cases that the authors claim te usability of the framework and if it is statistically insignificant in the cases where the authors claim the framework is not usable. Having the statistical tests will also allow to characterize the limits of the framework.

---

> ### Author Response · Authors · 2022-08-02
> **Reviewer GGdc**
>
> We thank the reviewer for the detailed response.
>
> Weaknesses:
> 1. We have made the suggested change in the updated manuscript, which was merely a confusing bit of notation and does not affect our results.
>
> 2. This observation is correct. In the updated version, we have included the following statement below Eq (7) to clarify this: “While the absolute magnitudes of this quantity are not informative on their own, the relative values can be compared to infer which of the two learning rules is more likely to have generated the data.”
>
> 3. The RNN simulations do not use a noise-averaged version of the RL weight update (Eq 3). We only use the noise averaged version of the RL weight update to motivate our proposed metric (Eqs 5-7) and to derive under what conditions the node-perturbed weight updates follow the gradient defined by the BMI decoder (Appendix B.2). We have clarified the language in the main text and appendix in our new version. The noise $\xi$ that we use for most simulations is relatively high, as can be seen from the noisy cursor trajectories in Figs 2A,D, and is necessary for RL exploration.
>
> 4. In general, roughly, the results are significant with p<0.05 whenever the shaded regions (SEM) in our plots are non-overlapping. To avoid clutter, we have chosen not to indicate statistical significance on every result that we present. However, we have explicitly computed statistical significance in the new Fig S8, which shows the effects of varying noise on our main results.
>
> Questions:
> 1. (Addressed in Weakness 1)
>
> 2. Each point in Figure 2F is the average of different instantiations of matrices $M$ and $W^{bmi1}$, where we randomly choose matrices $\hat{M}$ to have the desired alignment with $(W^{bmi1})^T$.
>
> 3. The flow field $F$ is a matrix that maps a point $h$ in neural activity space to another point in activity space. It is defined for all $h$, regardless of time. $\Delta F(h)$ is therefore the change in flow field, i.e. the difference between these two matrices, for any $h$.
>
> 4. We have fixed our notation in Eq 4 and the sentence following it.
>
> 5. We clarified in our revised text in line 169 that the relation between $\Delta F$ and $\Delta W$ only strictly holds when the other weight matrices are unchanged. We mention this as a challenge in line 316 of our updated version.
>
> 6. The feedback in our model is given by $y = W^{fb} W^{bmi} h$, and, as stated above, we assume that $W^{fb}$ and $W^{bmi}$ (determined by the experimenter) are fixed. When we perform the autoregression to compute $A$, part of this matrix will be a contribution from $W^{fb} W^{bmi}$, and that contribution will be the same for early and late trials, so that $\Delta F$, which subtracts these two quantities, will be unaffected. Thus, there are no necessary assumptions regarding $y$ beyond those already discussed above.
>
> 7. (Addressed in Weakness 2)
>
> 8.  We found that networks with alignment between M and $W^{bmi}$ of less than 0.3 mostly were unable to learn the cursor control task. On the other hand, our results show that learning rules become difficult to distinguish when the alignment is $\gtrsim 0.8$. This leaves a significant range of values in which our framework produces meaningful results.
>
> Limitations:
> 1. (See response to Weakness 2)
>
> 2. This was addressed in detail in our response to Reviewer SqGf. The short version is that (i) there aren’t many biologically plausible learning rules for training vanilla RNNs in the literature, and (ii) we have included an appendix on “biased BPTT” as an example to show that our framework applies to biased algorithms other than RFLO. Based on our theory and simulations, we think that there are good reasons to expect that biased vs. unbiased learning rules can be distinguished in general within our framework. We don’t mean to claim, though, that our approach would necessarily be capable of distinguishing more subtle differences between learning rules within each of these classes. Hence, our view is that two-way comparisons such as the ones we presented between biased and unbiased learning rules are more likely to lead to meaningful insights when applied to experimental data than trying to identify the one true learning rule from a large number of candidates.
>
> 3. Our RL rule is, in fact, a high-variance learning rule (Eq 3). This is likely why the correlation values for the RL-trained networks are much smaller than 1. The noise used for most simulations is relatively large, as can be seen from the noisy cursor trajectories in Figs 2A,D. This noise is necessary for RL exploration at each timestep and over trials.
>
> 4. As mentioned in the preceding response, the noise in our simulations was chosen to be fairly large in large part for this reason. We have also added Fig S8 to show how our main results change as the level of noise is increased still further. In general, we find that increasing noise causes our RNNs to no longer train successfully before it causes our flow field metric to fail.

---

> > ### Comment · Reviewer_GGdc · 2022-08-09
> > **Response to authors**
> >
> > I would like to thank the authors for responding to my comments. I would also like to apologize for the delay from my end in getting back to the authors.
> >
> > 1. **Scope of the work:** I appreciate the authors's clarification about their method being useful for distinguishing biased vs unbiased learning rules. I think this is a critical point in defining the scope of their work and I would appreciate if the authors could make this explicit in their abstract and introduction. Although the authors mention that their aim is to distinguish supervised vs reinforcement learning rules, I think it would be helpful to mention the biased vs unbiased point (I am willing to increase my score by 1 point if the authors make this change or agree to make this change).
> >
> > 2. **Noise in RL learning rule:** I apologize for my misunderstanding and thank the authors for clarifying this point. From the original submission, I failed to grasp this but this clarification was quite helpful. As mentioned above, this clarification is key to defining the scope of this work from the perspective of experimental validation of the proposed framework and if the authors are willing to update the presentation of their work, I am happy to increase my score to reflect my views.
> >     - As an additional clarification about the last point in limitations: I meant noise in measuring the neural activity, not the noise in the sense that the authors use, e.g. in the RL rule. But given the discussion about applying their framework to real world neuroscience data brought up by reviewer Jh97, I agree that this is beyond the scope of the current work.
> >
> > 3. **Concerns over the framework's rigour:** Although I believe the authors do a commendable job in presenting their motivation, their solution and demonstrating that their proposal is useful, I have my concerns over the mathematical details of the framework and I must admit that I found some of the authors' responses slightly hand-wavy.
> >     - First, the authors changed their notation from $h^t$ to $h$ while defining their flow although $h$ depends on $t$. I feel this slightly obscures a key detail and could be misleading to the reader. Specifically, the network state could be different at different time points and I think it is actually unreasonable to assume that the same $h$ can be obtained at any arbitrary time point. This point has other effects as discussed below.
> >     - The authors claim that the feedback is the same for early vs late stages of training. This assumption does not make sense to me. For the same input, the model output should be different after learning. Alternatively, no (effective) learning has happened in your model, i.e. the model has not improved in performing the task. The authors' assumption misrepresents this issue by dropping the $t$ subscript from the $h$ and assume that the same $h$ can exist in both early and late stages and the fact that the feedback and bmi weight matrices are held fixed, the feedback would also remain fixed. I believe that as the system learns, the state space statistics would also change and therefore, for the same input $x$ the trajectory of $h$ would be different. Consequently, the model output would be different and therefore the feedback would be different. I believe this is a core issue in computing the change in flow.
> >     - One possible way to mitigate this issue is to consider how different $h$ is for early vs late stages and it is possible that the authors' description of $\Delta F$ is a pretty good approximation to the true change in flow that can be measured. Ideally, the authors could characterize this approximation to make their proposal more rigourous. But given the time constraints of the rebuttal period (and to add to that my own delays), I believe it could be a tough ask.
> >     - My suggestion would be to address this point as a note to the readers while describing the mathematical details of the method (or add a sentence that this is an assumption for the method). Please feel free to refute me if I misunderstood something. If the authors agree to add this note, I am happy to increase my score to 6 because I believe that with some caution in mind, this work is a good contribution to the NeurIPS community.

---

> > > ### Author Response · Authors · 2022-08-09
> > > **Response to Reviewer**
> > >
> > > We thank the reviewer for their careful reading of our paper and their response, as well as for their positive overall assessment of our work and willingness to consider revising their previous score. We address the individual points below.
> > >
> > > 1. Scope of the work
> > >
> > > > I would appreciate if the authors could make this explicit in their abstract and introduction.
> > >
> > > We agree with the reviewer that this is an important point about the scope of our work that should – in addition to being discussed in our Discussion section – be made more clearly early in the paper. We will be happy to make this point more explicitly in the abstract and introduction of our final version.
> > >
> > > 2. Noise in RL learning rule
> > >
> > > > If the authors are willing to update the presentation of their work, I am happy to increase my score.
> > >
> > > As we mentioned in our response to Weakness 3 above, we have already made updates to the new text to clarify the misunderstanding about noise averaging of the RL update. Specifically, we have clarified the RL rule in the paragraph beginning on line 138, and clarified the flow field calculation in the paragraphs of our updated submission beginning with line 175. We have also updated the language around RL in Appendix B.2. We hope that this addresses the reviewer’s concern. If not, we would be happy to add additional clarifications about this that the reviewer thinks are necessary to the final version.
> > >
> > > Regarding the comment about measurement noise, the reviewer’s point seems to be that measurement noise and exploratory noise for RL are different things. In a sense, yes, since the latter is used for learning slowly over many trials. But this difference is irrelevant if one is asking about how noise corrupts observations of neural activity within each trial. Hence, if the reviewer is concerned about measurement noise, then we believe that the new Fig S8 is relevant for addressing this concern.
> > >
> > > 3. Concerns over the framework's rigour
> > >
> > > We believe that the reviewer’s concerns about our paper’s rigor are largely based on misunderstandings, which we attempt to address point by point below. We apologize for the misunderstanding and will make clarifications to our final version to avoid such confusions by other readers.
> > >
> > > > The network state could be different at different time points and I think it is actually unreasonable to assume that the same $h$ can be obtained at any arbitrary time point.
> > >
> > > We agree that this would be an unreasonable assumption, since, as the reviewer points out, the whole point of training the RNN is to change its activity trajectory in a useful way. However, this is not an assumption that we are making. Essentially, our flow field is a model, and $h^{n,t}$ are the data points used to train the model, while $h$ is a generic point at which we evaluate the trained model. If the reviewer’s point is that the model will give poor predictions if the training and testing points are in completely different parts of the space of possible $h$, then this is certainly true. However, our simulations (data not shown, though we would be happy to include this as a supplemental figure in the final version if the reviewer thinks it is important) show that this is generally not the case. During training, we find that RNNs tend to change their activity as little as possible while learning to perform a task correctly, so that the distribution of training and testing points is not drastically different. (Interestingly, this has also been shown to be the case for neural population activity in motor cortex in BMI experiments–see Golub et al. (2018).) Indeed, the fact that we obtain conclusive results from our simulated data gives evidence that the model we are fitting is doing a decent job.
> > >
> > > > The authors claim that the feedback is the same for early vs late stages of training. This assumption does not make sense to me.
> > >
> > > This is essentially the same misunderstanding as the previous point. Our assumption, which we point out explicitly in the manuscript, is that the feedback weights are not changed, not that the activity $h$ itself is the same early and late in training. As described above, $h^{n,t}$ are the data points used to train the model, while $h$ is a generic point at which we evaluate the trained model.
> > >
> > > > One possible way to mitigate this issue is to consider how different $h$ is for early vs late stages…
> > >
> > > As we mentioned above, we would be happy to include a supplemental figure on this in the final version. We apologize that we are not able to complete this in the few hours that the reviewer has left us with before the response deadline.
> > >
> > > > My suggestion would be to address this point as a note to the readers…
> > >
> > > We will be happy to add a clarification about the above points to the final version of our manuscript. We apologize for not being clearer about these points in our original submission.

---

> > > > ### Comment · Reviewer_GGdc · 2022-08-09
> > > > **Thank you for your clarifications -- increased my score**
> > > >
> > > > I would like to thank the authors for the clarifications they provided and agreeing to incorporate some of my suggestions. I am glad I could help improve the quality of this work.
> > > >
> > > > The points about noise is clear to me and given the author's updates during the rebuttal+discussion period, I believe my concerns are addressed.
> > > >
> > > > Thank you for elaborating on the early vs late hidden state distributions. I must admit I was not aware of Golub et al. (2018) and was not aware of this phenomenon that RNNs tend to change their activity very little while learning -- thank you for the pointer, I will read more about this to understand better. A supplementary figure about this would be very helpful.
> > > >
> > > > "We apologize that we are not able to complete this in the few hours that the reviewer has left us with before the response deadline." -- Apologies from my end for delay in responding to your comments. I do not expect the supplementary figure in the current version, but if the authors could add it in the camera-ready version, that would be great.
> > > >
> > > > Given these developments, I will update my score to 7. I wish the authors good luck and would like to congratulate them on their work.

---

### Official Review · Reviewer_Jh97 · 2022-06-29

**Rating:** 7
**Confidence:** 3
**Soundness:** 4 excellent
**Presentation:** 3 good
**Contribution:** 2 fair

**Summary:**

This paper introduces a novel metric for distinguishing biological learning rules from changes in neural activity.  The proposed metric is the correlation between the observed change in network activity and the predicted change in network activity for each learning rule under consideration.  Predictions are made by assuming that the brain can be modeled as a vanilla RNN, and by then deriving the expected change in neural activity for each learning rule of interest.  The authors show that their metric allows them to distinguish a biologically-plausible variant of backpropagation ("RFLO", Random Feedback Local Online) from the REINFORCE learning rule of Williams (1992) in simulation for a wide variety of parameter settings.

**Questions:**

My questions are mostly clarification questions:
- Linearity is assumed to get to equation 4.  Can you justify why this is a valid thing to do both in the simulated settings you consider, as well as how this assumption may or may not be valid when you move to real data.
- On line 170, you say that $M$ can be readily obtained from experimental data.  It was not obvious to me how you would do this.  My understanding is the $M$ is the matrix that the subject (e.g. monkey) $\it{thinks}$ maps neural activity to the cursor location, as opposed to the true readout matrix.

While answering these questions would help me better understand the paper, the biggest thing that would make this paper a clear accept for me would be a demonstration of the metric to real world data.

**Limitations:**

The authors were upfront about the limitations of their work, namely that:
- They only demonstrated the ability of their metric to distinguish two different learning rules (RFLO and REINFORCE)
- They only demonstrated the application of their metric in simulation

**Strengths And Weaknesses:**

Strengths:
- Clarity: this paper, with the few exceptions noted below, is well written. The figures were clear and easy to understand.
- Quality: the paper is rigorous and examines the usefulness of the proposed metric in a wide range of settings and in cases where the initial assumptions are invalid.  (As an aside: you might want to come up with a name for your metric!  The ML field loves an acronym and right now, I am not sure how best to reference it, other than "proposed metric".)
- Significance: the paper tackles an important problem in the computational neuroscience community, namely that of identifying biological learning rules from neural activity.  As noted by the authors (line 64), there is a lot of interest in proposing biologically plausible learning rules.  Having new ways to distinguish competing hypotheses is useful.
- Originality: to the best of my knowledge, this paper tackles an open-problem in the computational neuroscience community with a novel approach.

Weaknesses:
- Significance: the major limitation of this paper is the lack of application to real data.  The authors demonstrate the usefulness of their metric only when applying it to data simulated from an RNN.  Thus, it is hard to judge how useful their metric will be in real-world scenarios,  where some of the assumptions made by the authors may not hold.  For example, in lines 170-172, the authors say: "By averaging over noise, we effectively assume that the cumulative updates during learning that make up differ only in having different noise realizations, but are otherwise in a consistent direction."  Real world data is always messy, and I would be surprised if this assumption holds there.

Minor:
- Clarity: there's a typo in Equation 1, where the superscript on y is $t$, but I think it should actually be $t-1$ (as in Equation 15 in the supplement).  For equation 1, neither $\tau$ or $\phi$ are explicitly defined.  For reproducibility purposes, it would be useful for these quantities to be defined and for their values to be explicitly provided.

---

> ### Author Response · Authors · 2022-08-02
> **Response to Reviewer Jh97 - Applying our Analysis to Neuroscience Data**
>
> Thanks to the reviewer for their supportive review and insightful feedback.
>
> The single major criticism by the reviewer was that we should apply our theory to real neuroscience data. We absolutely agree, but we feel that this is beyond the scope of this paper for the following reasons:
>
> 1. Because the derivation and application of our metric to simulated data requires a significant amount of calculation and explanation, it is our view that the work presented here on its own represents a substantial contribution. A paper including data would necessarily have much more focus on experimental details and less on the development of the theory, to which we wanted to give due attention. Such a paper would have been more appropriate for a neuroscience journal rather than NeurIPS.
>
> 2. Our ideal experiment would be chronic recordings in M1 (~2 weeks) where experimenters change the decoder more than once. Such experiments take years to design and train monkeys on. Only a few research groups are capable of running such experiments, and we are hoping this simulation-based study will lead to collaborations with these groups. Such data is rare and extremely valuable. For example, by our count, one of the leading research groups in this field has published half a dozen papers in high-impact neuroscience journals using a single dataset of this type. The culture of openly sharing data has unfortunately not yet caught on in monkey neuroscience to the extent that it has in other fields. Publicly available monkey BMI data, to our knowledge, does not exist.
>
> The reviewer also made several constructive suggestions which gave us opportunities to clarify our presentation. Below are responses to the reviewer’s more-minor comments:
>
> >“There's a typo in Equation 1…”
>
> Thank you for catching these details. We have corrected the superscript on $y$ to $t-1$, define $\tau$ as the RNN time constant and $\phi$ as the activation function in the updated version. In the paragraph starting with line 537 of Appendix C.1 we state the $\tau$ value we used as 10 and the activation function $\phi$ as $\tanh$.
>
> >“Linearity is assumed to get to equation 4. Can you justify why this is a valid thing to do both in the simulated settings you consider, as well as how this assumption may or may not be valid when you move to real data.”
>
> It is important to emphasize that, although the task performed by our RNN is simple enough to be achieved by a linear network (cf. Fig. S4, which we have included to illustrate this), in which case our linearized theory would be fully justified, we chose to perform simulations in a nonlinear RNN for precisely the reason that the reviewer mentions, i.e. to model the nonlinearities in the brain and to show that our approach, despite the linear approximation, still enables us to draw conclusions about the learning rules used to train the RNN.
>
> >“On line 170, you say that M can be readily obtained from experimental data. It was not obvious to me how you would do this.”
>
> While we did propose and explain a procedure for this later on in the paper (Sec. 3.4 in the updated version), we failed to signpost the result appropriately where we made the claim in line 170. We have fixed this in the updated version.
> In Sec. 3.4, we propose one way to estimate M, and show that it works on our simulated data. This proposal requires learning two separate decoders, $W^{bmi0}$ and $W^{bmi1}$, and uses the observed neural activity during the learning of the first decoder to estimate the credit assignment mapping $M$. We then ask whether the network is using some estimate of $M$ during the learning of the second decoder.
>
> We think that this approach could plausibly be applied to real BMI experiments with non-human primates, by training the monkey on two separate decoders, and estimating a possible “credit assignment mapping” from the neural activity observed during the learning of the first decoder. An even simpler setup would be to equate $M$ with $(W^{bmi0})^\top$, instead of estimating $M$, and then analyze neural activity during the learning of $W^{bmi1}$.
>
> An ideal BMI experiment motivated by our framework would involve (i) chronic recordings in M1 before, during, and after BMI learning (ii) proficient learning of at least two decoders $W^{bmi0}$ and $W^{bmi1}$ and (iii) BMI decoder mappings that are difficult but learnable over multiple days. It is ideal to have at least two decoders so that two matrices can be compared - for example, when learning $W^{bmi1}$ after having proficiently learned $W^{bmi0}$, we can either estimate $M$ or equate $M$ with $(W^{bmi0})^\top$ and ask whether changes in neural activity lie in the subspace defined by the image of $M$ or in the subspace defined by the image of $(W^{bmi1})^\top$.
>
> Finally, we have named our proposed metric “FFCC” for Flow Field Change Correlation, and thank the reviewer for this suggestion.

---

> > ### Comment · Reviewer_Jh97 · 2022-08-09
> > **Response to authors**
> >
> > I thank the authors for responding to my review.  I am particularly grateful to them for clarifying the impracticality of applying their method to real data given the current climate of data-sharing in monkey neuroscience; I was not aware of this.  I will raise my score as my concerns have been alleviated.

---

### Official Review · Reviewer_VxtR · 2022-07-11

**Rating:** 7
**Confidence:** 4
**Soundness:** 3 good
**Presentation:** 4 excellent
**Contribution:** 3 good

**Summary:**

In this paper, the authors propose a method for distinguishing two biologically-relevant learning rules (unsupervised- and reinforcement-learning) based on neuronal activities in a recurrent neural network throughout an experiment similar to conventional studies with brain-machine interfaces. The authors verify their theoretical results in a series of simulated experiments where they confirm the model’s ability to distinguish the considered learning rules under various assumptions of different strengths.

**Questions:**

After reading the manuscript, I have a few minor questions remaining.

-the correlations of the observed and predicted changes in the flows (delta F observed and delta F predicted; Equation 5) are evaluated using the network’s states h visited in the experiment. Why? Is it because such states h are considered representative for the distribution of the possible states of the network, or is there another logic involved? Please clarify.

-Unless I’m missing something, in the model, all the network’s states h are considered observable and the only input x to the model is considered to be the task-relevant visual input. In the actual biological experiments, however, it’s hard to imagine recording from an entire functional circuit. Moreover, due to the unknown wiring of neurons on the individual-cell level, it may be hard to know what part of the functional circuit is being recorded. Therefore, it seems reasonable to assume that i) there’s an additional input to the network and ii) parts of the network’s activations are not observed. What modifications, if any, should be introduced to the proposed algorithm to account for such omission of the data?

-Although it’s outside the scope of this work, it would be great to see some estimate of how much data is needed to perform the proposed distinction on the biological data. How this volume of the data compares to the existing datasets and which ones can be used to run the proposed algorithm?

-Figure 3(E) caption: looks like it should say RL, not SL.

-The paper’s title is slightly confusing as there is no actual BMI data for now.


**Limitations:**

The limitations appear to be described/addressed fully. Overall, I think that this work is a thorough study of a problem highly relevant to the NeurIPS community.

**Strengths And Weaknesses:**

The goal of this paper is extremely worthy: whereas there’s plenty of theoretical work proposing various biologically plausible learning rules, little consideration is given to verifying these theories in practice using brain data. To address this issue, the authors here focus on the two prominent learning rules (i.e. the representative algorithms for supervised and reinforcement learning), for which they derive a theory and perform synthetic tests allowing distinguishing such rules based on typically-available data.

The paper has multiple strengths. The text is clearly written and well structured; the math is fully described in the appendix (requires no further reading!) and appears correct. The two considered learning approaches are representative of the learning algorithms, provided that the learning algorithms are often split into supervised, unsupervised, and reinforcement learning. The hypotheses in this work are introduced gradually (e.g. in the initial theory and simulations the credit assignment matrix M is considered known, while later on, this assumption is relaxed to “correlated with the real one”, and then to “learned from the data”). This approach streamlines the reading of the paper and also helps estimate the proposed method’s sensitivity to various parameters in the data. To this end, the authors propose the criteria of the method’s applicability.

---

> ### Author Response · Authors · 2022-08-02
> **Response to Reviewer VxtR**
>
> We thank the reviewer for their enthusiastic review and thoughtful comments. Below are responses to specific suggestions and questions:
>
> > “The correlations … are evaluated using the network’s states h visited in the experiment. Why?”
>
> When dealing with high dimensional neural activity space, we wanted to be careful about sampling from the part of the space where activity plausibly exists. As the reviewer points out, we considered such states h as representative for the distribution of the possible states of the network. We were careful to sample from non-overlapping subsets of points in neural activity space when generating predictions of the change in flow field for the matrices $M$ and $W^{bmi}$ (equations 5 and 6 of our revised submission), and when calculating the final correlation metric (equation 7). We have clarified our language around this the main text of our revised submission beginning with section 2.2 “Characterizing changes in neural activity with vector flow fields.”
>
> > “...it seems reasonable to assume that i) there’s an additional input to the network and ii) parts of the network’s activations are not observed. What modifications, if any, should be introduced to the proposed algorithm to account for such omission of the data?”
>
> The simulations in this study have assumed that the decoder $W^{bmi}$ reads out the neural activity from all the neurons in the RNN.  This assumption is not realistic with respect to neuroscience experiments, as the reviewer has pointed out. We completely agree with the reviewer on this point, and address it in our revised appendix. We ran simulations for RNNs where the observed neurons are only a subset of the full network, and find that it does not affect our conclusions.
>
> More specifically, we ran simulations for RNNs with 200 recurrent units using SL and varied the number of units read out by the decoder between 25 and 200, setting the decoder weights of all non-readout units to zero. Network and training hyperparameters were otherwise the same as in Figure 2. Figure S6 shows that, in this more realistic scenario where a BMI only decodes a subset of the neurons in the neural population, our correlation metric continues to distinguish between the SL and RL training algorithms.
>
> The reviewer also brought up the possibility that there is additional input to the circuit that is unknown and likely difficult to measure. We acknowledge that this is more difficult to take into account. In our modeling, we have made the assumption that the circuit inputs are the same throughout learning. In equation 1, this would correspond to variables $x$ and $y$ remaining roughly the same throughout learning.
>
> > “Although it’s outside the scope of this work, …how much data is needed to perform the proposed distinction on the biological data, and how does it compare to existing datasets?”
>
> Our modeling assumed access to 50 - 200 neurons, and between 500 - 1,500 trials for SL and 2,500-15,000 trials for RL. While the specific number of learning trials depends on the choices of hyperparameters, we think these numbers fall within the range of chronic BMI experiments.
>
> An ideal BMI experiment motivated by our framework would involve (i) chronic recordings in motor cortex before, during, and after BMI learning (ii) proficient learning of at least two decoders $W^{bmi0}$ and $W^{bmi1}$ and (iii) BMI decoder mappings that are difficult but learnable over multiple days.
>
> It is ideal to have at least two decoders so that two matrices can be compared - for example, when learning $W^{bmi1}$ after having proficiently learned $W^{bmi0}$, we can either (i) estimate $M$ using neural activity recorded during the learning of the first decoder or (ii) equate $M$ with $(W^{bmi0})^\top$, and ask whether changes in neural activity lie in the subspace defined by the image of $M$ or in the subspace defined by the image of $W^{bmi1}$. Finally, because there is the possibility that cerebellum is involved when learning “easy” BMI mappings and perturbations, it would be ideal to make the decoder sufficiently difficult that it requires multi-day learning, presumably involving long-term plasticity in motor cortex.
>
> > "Figure 3(E) caption"
>
> Thank you for catching the typo in the caption of Figure 3E; we have corrected it in the updated version.

---

> > ### Comment · Reviewer_VxtR · 2022-08-03
> > **Re: response**
> >
> > Thanks for the thorough response and for the provided clarifications.
> >
> > I especially appreciate the additional analysis performed by the authors that shows the method's capability to distinguish the two learning rules while observing partial activities of the network's neurons. I think this result makes the proposed method more ready to face the challenges of real-world BMI data.
> >
> > As to whether the real-world experiments should be included in this paper, I tend to agree with the Authors' arguments that: 1) the BMI data may be difficult to obtain and 2) that, if such experiments were included, the paper would've been more relevant to neuroscience journals, whose readers are likely less interested in the model. For these reasons, I believe that the model alone is a sufficient contribution to NeurIPS.

---

### Official Review · Reviewer_SqGf · 2022-07-11

**Rating:** 6
**Confidence:** 4
**Soundness:** 3 good
**Presentation:** 2 fair
**Contribution:** 2 fair

**Summary:**

This paper presents a method for distinguishing learning rules in the brain using data observed via brain-machine interface (BMI) experiments. The attempt to distinguish learning rules is motivated by the well-known fact that backpropagation is biologically implausible due to (potentially among other reasons) the weight-transport problem.

The weight transport problem is especially acute when fitting to data from BMI experiments because the weights mapping neural activity read by the BMI to behavior exhibited by the subject may change abruptly and cannot be immediately estimated for global error propagation. This approach therefore requires setting up a machine learning experiment that avoids the weight transport problem.

The authors model a BMI task using a vanilla recurrent neural network (RNN) with input weights W_in, recurrent weights W_rec, feedback weights W_fb, and BMI decoder weights W_bmi. W_bmi is fixed and learning only occurs in W_rec. In one set of experiments, the RNN is trained with supervised learning, specifically the Random Feedback Local Online (RFLO) rule that avoids the weight-transport problem by assuming that the weights carry an imperfect approximation of the ideal credit assignment mapping and dropping local terms from the weight update. This is expected to be a biased estimator. In the other experiments, the RNN is trained with reinforcement learning with node perturbation and subtracting off a baseline, yielding a somewhat noisy but unbiased estimator. Running these training processes yields a synthetic version of the data that would come from a BMI.

The two different learning paradigms are distinguished via a metric based on vector field flows over training. The authors hypothesize that supervised learning will have bias emerge over the course of training because of RFLO while reinforcement learning will not, making them distinguishable. They present the results of the training performance and distinguishability via flow field-based metrics.


**Questions:**

-	Why does the weight matrix mapping neural activity onto cursor position change abruptly?
-	How much does the systematic bias claim extend to anything beyond RFLO vs. node-perturbed RL?
-	How does the experimenter know the decoder? In general, more background on BMIs would be useful – not because I doubt the claims, but because it would help to have a better mental model.
-	How is the assumption about partial alignment with W_bmi^T justified? For those not deeply familiar with RFLO, could you expand a little bit on this detail?


**Limitations:**

The experimental and ethical limitations are stated sufficiently. I consider there to be more limitations, but these are likely the products of a difference of opinion between myself and the authors. The fact that they aren’t here is not a knock on the quality of this section.

**Strengths And Weaknesses:**

Edit: author comment + context from other reviews is more convincing of the value of this experiment for the field. Raising score.

**Strengths**
*Originality*: This paper takes the well-known problem of understanding and modeling biological learning rules and the existing tool, brain-machine interfaces, and creates a novel application. This application requires creative training formulations and metrics to test and evaluate. To my knowledge, this is novel – the problem and tool exist, but in different fields. The application is non-trivial and unique. The authors show a representative sample of related prior work regarding investigating biological learning rules from experimental data, including the most similar studies I know of (Lim et al. and Nayebi et al.) and distinguish from them. Furthermore, they present some related work on modeling BMI experiments with RNNs; I am not aware of any BMI work that is more related than this.

*Quality*: This paper justifies all its small claims (i.e. details) correctly. The experiment setup with supervised learning and reinforcement learning seems correct to my understanding, the hypothesis about bias specifically from RFLO and noise without bias from node-perturbation-based RL are reasonable, results from execution make sense, and the metrics are presented and used to validate the approach convincingly.

In my opinion, the paper does not deliver on larger claims in a cascading way. First, the central hypothesis explaining the distinguishability seems to be that supervised learning (not just RFLO) will show bias emerge in training trajectories and reinforcement learning won’t. However, only one local supervised learning rule is shown and the explanation for bias emerging is specific to it. The related works cited here present other local supervised learning rules; they should at least be discussed to make a claim about supervised learning that avoids the weight-transport problem. This then cascades into a larger problem – being able to distinguish between some local supervised learning and some reinforcement learning doesn’t support the larger claim that this method can distinguish between learning rules, at least not in my opinion of the spirit of this claim. Finally, even if we assume these claims are supported, this cascades into the fact that we don’t know enough about biological learning rules to know that ability to distinguish between local supervised learning and reinforcement learning will be highly productive; there is no biological experiment to support this. The authors cite some related work about biological plausibility of each of these, but basing claims from purely simulated experiments on these fit experiments seems insufficient.

Even if others disagree with me on the spirit of the claim the paper makes in its title, intro, and discussion, ultimately we have seen that simulated BMI experiments with flow field-based metrics can distinguish between (an example of) biased and unbiased learning trajectories. I consider this far from the scope of the main claims.

*Clarity*: The writing in this paper is clear and useful for understanindg. I didn’t notice any concerning typos, and the ideas get across. The clarity suffers somewhat on the structure and figures. For structure, the paper basically has two qualitative sections (intro and conclusion, total ~2 pages) and one very long section (termed “results”) with everything else in it; the results only arrive in the second half, after methods have all been communicated. At the moment, the fact that “Results” are signposted before the reader knows experiment setup, the fact that the results don’t come until a long time after “Results” has begun, and the general lack of structure make this paper confusing and a bit overwhelming to read. It would be as simple as breaking this section into “Methods” and a couple results sections (e.g. one for performance, one for distinguishability) to fix that.

The figures contain the right information, but also aren’t clearly presented. Most of the figures contain many different subfigures, which is fine, but they aren’t well-annotated – it is helpful to show significant points on the charts directly. Furthermore, schematic parts of the figures (like all of figure 1) could benefit from less visual detail in the icons – lots of squiggly lines may look nice and even be more faithful to reality, but if they don’t improve our understanding of the point of the figure, they aren’t necessary and can be visually cluttered.

*Significance*: As stated in the quality section, the claims of this paper are big and worthwhile. The setup and idea are also completely worthwhile: the idea of doing ML-analogous real experiments with BMIs is promising and this paper gives a good example of setting such a direction up. The fact that this setup led to conclusive experiments about learning rules may be significant enough to warrant publication (I’m unsure). But not only do the results not live up to the broad claims of the paper, they don’t make a very significant *result contribution* for machine learning for the reasons discussed in the quality section. We know simulated BMI experiments are a promising tool because of this paper, but not that they will provide publication-level value. Combined with the fact that this promise hasn’t been validated in real BMI experiments, I feel that we end up without a real “get” in this paper.

---

> ### Author Response · Authors · 2022-08-02
> **Reviewer SqGf**
>
> The reviewer’s major concern is that, while our approach enables us to distinguish between the two learning rules that we consider, we are not able to definitively distinguish between _classes_ of learning rules. We fully agree that this is a limitation of our work and discussed it in some detail in the second paragraph of our Discussion section. The point that we attempted to make there is that any possible learning rule either does or does not make use of a credit assignment mapping. If the algorithm does make use of such a mapping, then, given that there is no plausible way for the brain to instantly have perfect information about how its neural activity maps onto behavior, this mapping will necessarily be biased, and this bias will leave a signature in the neural activity. If future researchers have different candidate learning rules that they would like to test, they will be able to use our framework for those learning rules. Though we don’t have a fully general proof, we conjecture that our approach will be capable of distinguishing biased from unbiased learning rules with a fair degree of generality (cf. our discussion below on Appendix C.2, where we apply our approach to distinguish biased vs. unbiased versions of BPTT).
>
> Relatedly, the reviewer seems to be concerned that there are a large number of biologically plausible learning rules in the literature for training RNNs, and that we are cherry picking. From a systematic review of approximate gradient-based learning rules for vanilla RNNs (Marschall et al, 2020), RFLO is the only one that is fully local, and hence, according to our criteria, biologically plausible. In the last two years, the most prominent biologically plausible algorithm for training RNNs has been e-Prop (Bellec et al, 2020), which is essentially a generalization of RFLO to spiking networks. For RL, the only other algorithm that we are aware of besides the simple node perturbation that we use is from Miconi (2017), which is so similar that it would be highly unlikely to change our main results.
>
> While the reviewer’s assessment that “The setup and idea are also completely worthwhile…The fact that this setup led to conclusive experiments about learning rules may be significant enough to warrant publication” is encouraging, we aren’t sure what changes would satisfy the reviewer’s main criticism. We don’t believe that a complete proof for all possible learning rules is attainable, and, while we would be happy to perform additional simulations using extra learning rules, it isn’t clear how useful this would be since there aren’t many others in the literature for vanilla RNNs, and the ones that are in the literature are minor variations on the ones that we consider and are unlikely to give different results. If the reviewer feels that our work is satisfactory but our claims are unsupported, then we would consider amending our language, but have already tried to be careful about this in the paper and discussed it appropriately as a limitation, so we would appreciate more concrete suggestions about which specific changes the reviewer would like to see.
>
> >“We don’t know enough about biological learning rules to...be highly productive...”
>
> We interpret the reviewer's point to be that the brain may be using very different learning mechanisms than the ones that we consider, in which case our results would have limited relevance to neuroscience. In fact, there is a substantial amount of experimental evidence for so-called 3-factor learning rules in the brain, in which plasticity depends on a multiplicative combination of pre- and postsynaptic activity, as well as a third factor that contains information about error or reward. The learning rules that we consider fall within this framework, and we have added a citation in line 497. Thanks for helping us to clarify this important fact.
>
> We have divided our Results section into separate Theory and Simulation results sections.
>
> Q1,Q3: The experimenter knows the decoder because they get to define it, and abruptly changing it is a standard feature of BMI experiments, creating a learning problem that the experimental subject has to solve. We have added minor clarifications in the paragraph at line 41 to make these points clearer.
>
> Q2: In Appendix C.2, we show that our systematic bias claim extends to biased and unbiased versions of BPTT. Although BPTT is nonlocal, this shows that we are able to distinguish between biased and unbiased algorithms other than RFLO and node-perturbation RL.
>
> Q4: This idea is not specific to RFLO and applies to gradient-based algorithms generally. A supervised algorithm can only learn if there is positive alignment between the readout and feedback weights (Lillicrap et al, 2016). Thus, the fact that monkeys are able to learn BMI tasks means that, if they are using SL, there must be positive alignment between these matrices (presumably due to learning of $M$, since $W^{bmi}$ is fixed).

---

> > ### Comment · Reviewer_SqGf · 2022-08-09
> > **Re: response**
> >
> > Thanks to the authors for the detailed response!
> >
> > My main concern, as you sussed out, is that the paper felt incremental relative to claims. This response clarifies the context behind some of the decision making and experimental design and I buy that it's well-motivated and a useful contribution. I especially appreciate the context about three-factor learning rules, since my knowledge about this space is likely somewhat outdated.
> >
> > In terms of language, I do still feel like statements such as "method for distinguishing biased SL from unbiased RL under the assumption that the mapping from the brain to behavior is known" overstates what we learn about the brain from these experiments (we know it's supervised learning?) but on re-read I do agree that the majority of the paper is careful about the specific insights. So I'm not going to push on this.
> >
> > Score raised.

---

### Meta-Review · Area_Chair_2Mhi · 2022-08-23

**Recommendation:** Accept
**Confidence:** Certain

**Metareview:**

This paper explores the question of experimentally distinguishing between different hypothesized classes of learning rules in the brain (specifically biased supervised learning and unbiased reinforcement learning). It derives a metric to distinguish between such learning rules based on changes in neural activity seen during learning with a brain-computer interface. The authors show that this metric can be used to identify which learning rules are the best account of the observed activity changes.

The reviewers agreed that this paper makes an original and important contribution to the field, and the decision to accept was unanimous.



**Award:**

No

---

### Decision · Program_Chairs · 2022-09-14

Accept